# High-fidelity collisional quantum gates with fermionic atoms

Petar Bojović[1,2], Timon Hilker[1,2,3], Si Wang[1,2], Johannes Obermeyer[1,2], Marnix Barendregt[1,2], Dorothee Tell[1,2], Thomas Chalopin[1,2,4], Philipp M. Preiss[1,2], Immanuel Bloch[1,2,5] & Titus Franz[1,2 ✉]

Quantum simulations of electronic structure and strongly correlated quantum phases are among the most promising applications of quantum computing. These computations benefit from native fermionic encodings[1,2], enforcing fermionic statistics and conservation laws such as particle number and magnetization[3] independent of gate errors. While ultracold atoms in optical lattices have become established as powerful analogue simulators of strongly correlated fermionic matter[4–7], neutral-atom platforms have concurrently emerged as versatile, scalable architectures for spin-based digital quantum computation[8]. Unifying these capabilities requires high-fidelity motionally coherent gates for fermionic atoms[9–11], similar to collisional gates in bosonic systems[12,13], paving the way for programmable fermionic quantum processors. Here we demonstrate collisional entangling gates with fidelities up to 99.75(6)% and Bell-state lifetimes exceeding 10 s, realized by means of controlled interactions of fermionic atoms in an optical superlattice. Using quantum gas microscopy[14], we microscopically characterize spin-exchange and pair-tunnelling gates and realize a robust composite pair-exchange gate, a key building block for quantum chemistry simulations[3,15]. Our results establish controlled collisions in optical lattices as a competitive and complementary route to high entangling gate fidelities in neutral-atom quantum computers. Operating intrinsically with fermions, this capability naturally extends to many-qubit architectures, in which fermionic statistics become relevant, enabling complex state preparation and advanced readout[16–19] in scalable analogue–digital hybrid quantum simulators. Combined with local addressing[20,21], these gates mark a crucial step towards a fully digital fermionic quantum computer based on controlled motion and entanglement of neutral atoms.

Neutral atoms have proved to be a compelling platform for quantum simulation[22–24] and quantum computing[8], offering long lifetimes, inherent scalability, strong tunable interactions and naturally identical qubits[8,19,25]. In the context of spin-based quantum computing, most of the progress so far has been focused on implementations using Rydberg interactions to realize fast and robust quantum gates[26–30]. However, collisional quantum gates were proposed early on as a promising alternative, presenting the potential for high-fidelity operations in both spin and charge degrees of freedom[31–34]. Whereas first foundational experiments showed the merit of such collisional gates using effective spin-1/2 degrees of freedom encoded in bosonic atoms[35–38], they lacked single-site and single-spin-sensitive readout capabilities and were thereby limited in the microscopic assessment of realized gate fidelities. With the advent of quantum gas microscopy, a better characterization of such collisional quantum gates became possible, including the realization of complex multiparticle entangled quantum states[39].

The highest reported two-particle entangling gate fidelities from such microscopic analysis reached approximately $F \approx 96\%$ (ref. 39), whereas earlier experiments without microscopic resolution reported fidelities as high as $F \approx 99.3\%$ (ref. 13).

In this work, we demonstrate entangling collisional gates using fermionic $^6$Li atoms in both spin and charge degrees of freedom—essential building blocks for quantum computing architectures based on fermions. Using quantum gas microscopy, we determine both the continuous-time and discrete gate-based performance of the collisional interactions. For the latter, we find high entangling gate fidelities up to 99.75(6)%. Furthermore, we generate entangled Bell states and observe noise-resilient entanglement with lifetimes exceeding 10 s. We find that the main source of gate infidelity originates from averaging over double-wells with slightly different oscillation frequencies, indicating strong prospects for achieving even higher fidelities using, for example, echo pulses, optical potential flattening procedures[40–42] and optimal control sequences[43,44]. Furthermore, we engineered a pulse sequence

[1]Max-Planck-Institut für Quantenoptik, Garching, Germany. [2]Munich Center for Quantum Science and Technology, Munich, Germany. [3]Department of Physics and SUPA, University of Strathclyde, Glasgow, UK. [4]Laboratoire Charles Fabry, Institut d'Optique Graduate School, CNRS, Université Paris-Saclay, Palaiseau, France. [5]Fakultät für Physik, Ludwig-Maximilians-Universität München, Munich, Germany. ✉e-mail: titus.franz@mpq.mpg.de

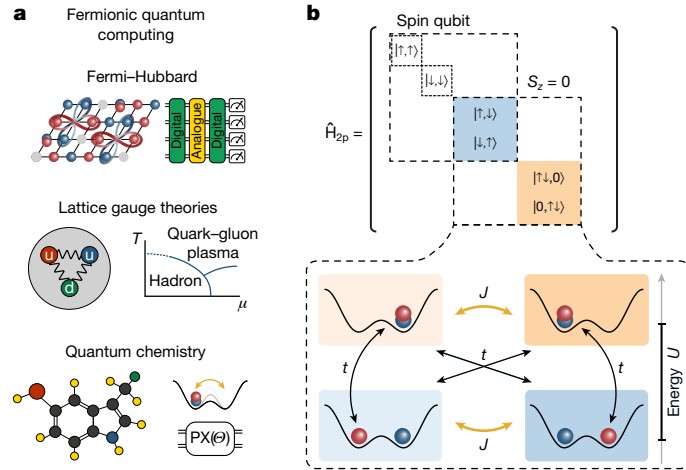

**a** Fermionic quantum computing

Fermi–Hubbard

Lattice gauge theories

Quark–gluon plasma

Quantum chemistry

**b** Spin qubit

Energy $U$

**Fig. 1 | Fermionic quantum processor. a**, By controlling the spin and motional dynamics of fermionic atoms with digital gates, a future lattice-based quantum processor can efficiently simulate strongly correlated systems of many electrons or other fermionic particles. **b**, At its core, double-well potentials of an optical superlattice are used to entangle particles. In the two-particle sector of the Fermi–Hubbard Hamiltonian, the single-particle tunnelling processes $t$, detuned by the on-site interaction $U$, lead to the spin-exchange and pair-tunnelling, both characterized by an effective coupling $J$. Through appropriate experimental control sequences, the resulting dynamics realize high-fidelity SWAP$^\alpha$ gates.

that decouples spin-exchange from pair-tunnelling dynamics, thereby enabling a pair-exchange gate. In conjunction with single-particle hopping[14,16], this mechanism is an essential building block for future digital fermionic quantum computers with applications in materials science and chemistry[2,3,45,46] (Fig. 1a, bottom). Such a future device encodes fermionic problems more directly, thus naturally restricting the Hilbert space to the one for fermionic particles, thereby intrinsically preventing any non-physical states. Also, it conserves particle number as well as polarization independent of gate errors. In the near term, these gates already enable new readout and preparation schemes of strongly correlated electronic states[17–19] in hybrid analogue–digital quantum simulation schemes (Fig. 1a, top). In the long term, fermion-based architectures also hold the potential to simulate the dynamics of lattice gauge theories[2,47] (Fig. 1a, middle).

The dynamics of spin-1/2 fermions confined in a double-well potential is captured by the two-site (L, R) Fermi–Hubbard Hamiltonian

$$
\widehat{H}_{\mathrm{FH}} = -t \sum_{\sigma} (\hat{c}^{\dagger}_{\mathrm{L},\sigma} \hat{c}_{\mathrm{R},\sigma} + \mathrm{h.c.}) + U \sum_{i} \hat{n}_{i,\uparrow} \hat{n}_{i,\downarrow}
$$
$$
+ \frac{\delta}{2} \sum_{\sigma} (\hat{n}_{\mathrm{R},\sigma} - \hat{n}_{\mathrm{L},\sigma}) + \Delta_{B}(\hat{n}_{\mathrm{R},\uparrow} - \hat{n}_{\mathrm{R},\downarrow}) \tag{1}
$$

in which $t$ is the particle tunnelling energy, $U$ the on-site repulsive interaction and $\delta$ indicates a spin-independent potential offset between the left and right sites of the double-well. A spin-dependent chemical potential gradient, induced by an applied magnetic field gradient, is denoted by $\Delta_{B}$. The operator $\hat{c}_{i,\sigma}$ ($\hat{c}^{\dagger}_{i,\sigma}$) annihilates (creates) a fermion with spin $\sigma \in \{\uparrow,\downarrow\}$ on site $i \in \{\mathrm{L}, \mathrm{R}\}$ and $\hat{n}_{i,\sigma}$ represents the corresponding density operator. In the minimal configuration relevant for collisional gates—two fermions in the double-well—the Hamiltonian $\widehat{H}_{\mathrm{FH}}$ effectively reduces to a six-dimensional Hilbert space. Within this space, $\widehat{H}_{\mathrm{FH}}$ acts non-trivially only on the four-dimensional subspace with total spin projection $S_z = 0$, in which spin-exchange and correlated pair-tunnelling naturally emerge as second-order processes with identical exchange amplitude $J$ (Fig. 1b; for the general expression, see the Supplementary Information).

Full control over the motional degrees of freedom of two fermions in a double-well is a key prerequisite for digital fermionic quantum computation, requiring the ability to isolate individual tunnelling from exchange processes. This is achieved experimentally by implementing a unitary interaction matrix in the basis $\{|\uparrow,\uparrow\rangle, |\downarrow,\downarrow\rangle, |\uparrow,\downarrow\rangle, |\downarrow,\uparrow\rangle, |\uparrow\downarrow,0\rangle, |0,\uparrow\downarrow\rangle\}$:

$$
U_{\mathrm{int}}(\theta) =
\begin{bmatrix}
1 & 0 & 0 & 0 & 0 & 0 \\
0 & 1 & 0 & 0 & 0 & 0 \\
0 & 0 & \dfrac{1+e^{i\theta}}{2} & \dfrac{1-e^{i\theta}}{2} & 0 & 0 \\
0 & 0 & \dfrac{1-e^{i\theta}}{2} & \dfrac{1+e^{i\theta}}{2} & 0 & 0 \\
0 & 0 & 0 & 0 & e^{-i\zeta}\dfrac{1+e^{-i\theta}}{2} & -e^{-i\zeta}\dfrac{1-e^{-i\theta}}{2} \\
0 & 0 & 0 & 0 & -e^{-i\zeta}\dfrac{1-e^{-i\theta}}{2} & e^{-i\zeta}\dfrac{1+e^{-i\theta}}{2}
\end{bmatrix},
\tag{2}
$$

in which the angle $\theta$ is determined by the exchange coupling $J$ and the gate duration $\tau_{\mathrm{h}}$, whereas the angle $\zeta$ depends on both the on-site interaction $U$ and $\tau_{\mathrm{h}}$. For a quench, $\theta = 2\pi J\tau_{\mathrm{h}}/h$ and $\zeta = 2\pi U\tau_{\mathrm{h}}/h$, in which $h$ is Planck's constant (for a detailed derivation, see the Supplementary Information).

The upper-diagonal $4 \times 4$ block captures the unitary dynamics of the spin degree of freedom, effectively realizing a SWAP$^{\alpha}$ ($\alpha = \theta/\pi$) gate within this subspace. For $\alpha = 1/2$, this implements the entangling $\sqrt{\mathrm{SWAP}}$ gate, which, together with single-qubit gates, forms a universal gate set for spin-based quantum computers[48,49]. The lower-diagonal $2 \times 2$ matrix describes coherent pair-tunnelling, which entangles two particles in their charge degrees of freedom.

In our experiment, we load a Fermi-degenerate gas of $^6$Li atoms into a two-dimensional square optical lattice to prepare a state with two atoms of opposite spins at most lattice sites[7]. Individual lattice sites are split into double-wells by superimposing a lattice in the $x$-direction with exactly half the lattice spacing (short lattice), forming an optical superlattice. Lattice depths are expressed in units of their respective recoil energy $E_{\mathrm{r}} = h^2/8ma^2$, with atomic mass $m$ and respective lattice constant $a$. By setting the double-well bias $\delta = 0$, a symmetric double-well potential is formed, resulting in the splitting of doublons into dimer singlets[11,14] (Fig. 2a). Similarly, we can prepare the highlighted eigenstates in Fig. 1b: to initialize a product state $|\Psi_0\rangle = |\uparrow,\downarrow\rangle$, we apply a strong magnetic field gradient along the $x$-direction during splitting while maintaining the symmetric configuration[50] (Fig. 2b, upper plot). The states $|\uparrow\downarrow,0\rangle$ and $|0,\uparrow\downarrow\rangle$ are initialized by introducing a controlled offset $\delta$ while ramping up the short lattice (Fig. 2b, lower plot)[36]. After a quantum gate is applied (Fig. 2c), we perform a projective measurement and detect the final state using spin and charge site-resolved fluorescence imaging[51].

We first probe continuous spin-exchange and pair-tunnelling dynamics. The spin-exchange interaction, which couples the states $|\uparrow,\downarrow\rangle$ and $|\downarrow,\uparrow\rangle$, is initiated by lowering the intra-double-well barrier in 500 μs. The resulting dynamics are visualized on a two-particle Bloch sphere (Fig. 3a, top) as a rotation around the $x$-axis in which the entangled state $\frac{1}{\sqrt{2}}(|\uparrow,\downarrow\rangle - \mathrm{i}|\downarrow,\uparrow\rangle)$ is populated after a $\theta = \pi/2$ rotation. For our parameters (Methods), we observe long-lived oscillations of the population $|\uparrow,\downarrow\rangle$ with a frequency of $J/h = 3.32(3)$ kHz (Fig. 3b) on 20 central lattice sites. From the Gaussian decay of the contrast of these oscillations $\propto e^{-(\tau_{\mathrm{h}}/\tau_{\mathrm{ex}})^2}$ (Fig. 3c, bottom), we extract a $1/e$ decay time of $\tau_{\mathrm{ex}} = 33(2)$ ms, corresponding to a quality factor of $J\tau_{\mathrm{ex}}/h = 110(8)$ coherent oscillations—the highest recorded in superlattice platforms[13] and for any collisional entangling gates[9,12]. A related two-particle exchange dynamics can be realized in the charge degree of freedom

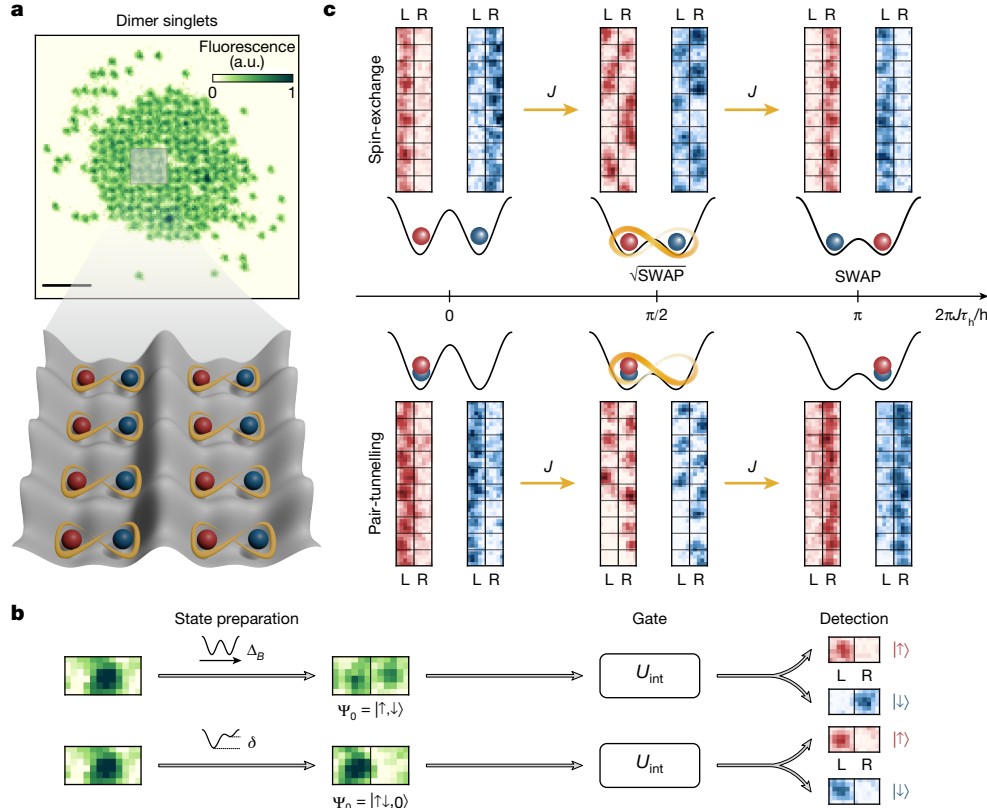

**Fig. 2 | Two-particle gates in double-wells. a**, Example shot of dimer singlets in a double-well lattice. **b**, Experimental sequence for Néel or doublon initial states, which are prepared by applying a spin-dependent chemical potential $\Delta_B$ (top) or a double-well tilt $\delta$ (bottom). **c**, Experimental shots of a $2 \times 10$ subsystem showing continuous SWAP$^\alpha$ evolution from the initial product state (left) through the entangled state (middle) to the swapped product state (right) in the spin (top) and doublon (bottom) sectors. a.u., arbitrary units. Scale bar, 5 μm.

when the system is initialized in the state $|\uparrow\downarrow,0\rangle$. This state is coupled to $|0,\uparrow\downarrow\rangle$ (see Bloch sphere in Fig. 3a, bottom), leading to coherent pair-tunnelling[36,52]. Constraining the analysis to a region of ten lattice sites and post-selecting on measurements with only these two states, we extract a 1/e decay time of $\tau_{ex} = 15(3)$ ms, corresponding to a record-high $J\tau_{ex}/h = 55(10)$ coherent oscillations in the population of $|\uparrow\downarrow,0\rangle$ (Fig. 3d,e).

To assess the effect of spatial inhomogeneity on the observed coherences, we extract maps of the local gate oscillation frequency (Fig. 3c,e, upper panels). We find that the spatial variation in frequency is consistent with the observed decoherence time, indicating that, in both cases, spatial inhomogeneity is the dominant limitation of the quality factor (Methods). The essential distinction between spin-exchange and pair-exchange lies in the increased sensitivity of the latter to motional dynamics: any tilt $\delta$ in the double-well potential, whether caused by phase shifts between the short and long lattices or by lattice beam inhomogeneities, lifts the degeneracy between the states $|\uparrow\downarrow,0\rangle$ and $|0,\uparrow\downarrow\rangle$. This results in dephasing that scales as $(\delta/J)^2$, in contrast to the $(\delta/U)^2$ scaling characteristic of spin-exchange. The heightened sensitivity accounts for both the pronounced spatial dependence of the pair-exchange frequencies and the more rapid decay of contrast (Fig. 3c,e, lower panels).

Coherent spin-exchange interactions in a double-well potential can be used to realize two-qubit SWAP$^\alpha$ gates, as shown in equation (2), with the relevant Hilbert space formed by $\{|\uparrow,\uparrow\rangle, |\downarrow,\downarrow\rangle, |\uparrow,\downarrow\rangle, |\downarrow,\uparrow\rangle\}$. To implement this in the experiment, we need to suppress single-particle tunnelling events that mix the spin sector $\{|\uparrow,\downarrow\rangle, |\downarrow,\uparrow\rangle\}$ with the charge sector $\{|\uparrow\downarrow,0\rangle, |0,\uparrow\downarrow\rangle\}$. This can be achieved in the regime $U/t \gg 1$, in which such processes become far off-resonant at the cost of reducing gate speed $J \approx 4t^2/U$. An approach leading to faster gate operation is to

fine-tune $U/t$ to a magic ratio $4/\sqrt{3}$ (ref. 13) (Supplementary Information). For this ratio, the effective single-particle tunnelling rate is exactly four times larger than $J$, leading to decoupling of the spin and charge sectors at integer multiples of $\tau_h = h/(4J)$. This is confirmed in our measurements (Fig. 4a, left), in which we quench the dynamics by ramping down the short lattice depth over 50 μs, a duration that is fast compared with the inverse of the energy gap $U$, thereby reaching a regime with $U/t = 4/\sqrt{3}$. We observe a substantial fraction of up to 40% of doublons excited during the time evolution $\tau_h$, which decreases to less than 5% for a $\theta = \pi/2$ entangling $\sqrt{\text{SWAP}}$ pulse (dashed lines) or $\theta = \pi$ spin-exchange SWAP pulse (dotted lines)[13]. However, the required fine-tuning of the gate parameters can make it less robust to technical imperfections.

In an alternative approach, we introduce an intermediate-speed strategy in which the tunnelling $t$ is ramped slowly compared with $U$ but still fast with respect to $J$. This avoids mixing the spin and charge sectors without the need to keep $U/t \gg 1$ or to fine-tune $U/t$ (Supplementary Information). In this case, the total doublon fraction never exceeds 8%, independent of gate duration (Fig. 4a, bottom middle). Stronger suppression of doublon excitations to less than 5% can be achieved by using smoothly shaped Blackman pulses longer than 0.75 ms (Fig. 4a, right). We ramp the short lattice depth from $54\,E_r^{short}$ to $6.50\,E_r^{short}$, reaching a minimum of $U/t \simeq 2.85$, with the long lattice depth held constant at $38.0\,E_r^{long}$. As a result of the high intrinsic energy scales in our system, our $\sqrt{\text{SWAP}}$ gate duration of 1.125 ms (dashed line) remains notably faster than previously reported durations[13,35–38]. This quasi-adiabatic approach based on Blackman pulses provides a good compromise between robustness and speed. Hence, we use it in the remainder of this section. Figure 4b shows the truth table for a single $\sqrt{\text{SWAP}}$ pulse applied to $|\uparrow,\downarrow\rangle$ and $|\downarrow,\uparrow\rangle$. The spin-polarized states $|\uparrow,\uparrow\rangle$ and $|\downarrow,\downarrow\rangle$ are not coupled to any other state by

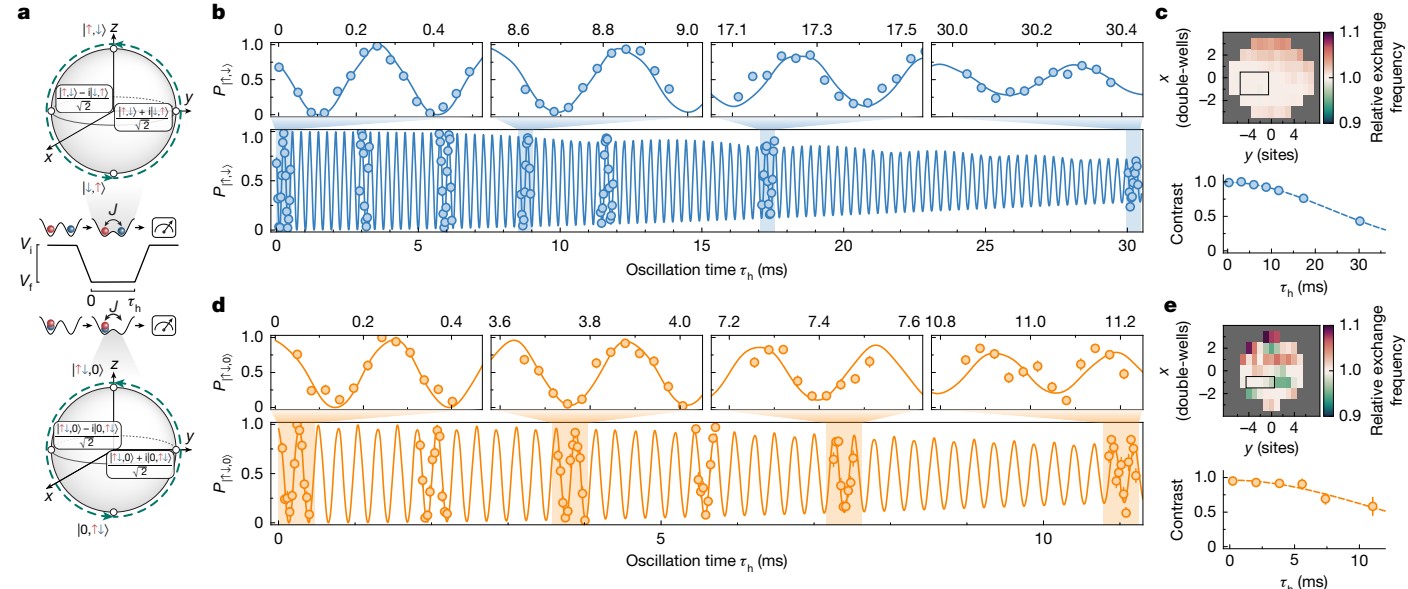

**Fig. 3 | Coherent spin-exchange and pair-tunnelling. a**, Evolution of the initial state $|\uparrow,\downarrow\rangle$ (top) and $|\uparrow\downarrow,0\rangle$ (bottom) on a two-particle Bloch sphere under exchange $J$ controlled by the lattice depths (middle). **b**,**d**, The populations of the states $|\uparrow,\downarrow\rangle$ and $|\uparrow\downarrow,0\rangle$, each evolving under exchange interactions with frequencies $J/h = 3.32(3)$ kHz and $J/h = 3.8(2)$ kHz, respectively. The outsets show enlarged sections of the full dataset. The data are post-selected on double-wells that only contain $|\uparrow,\downarrow\rangle$ and $|\downarrow,\uparrow\rangle$ (in **b**) and $|\uparrow\downarrow,0\rangle$ and $|0,\uparrow\downarrow\rangle$ (in **d**). We find good agreement with numerical simulations of the Fermi–Hubbard

Hamiltonian in equation (1) (solid lines), which exhibit a small anharmonicity owing to a finite ramp speed. **c**, Top, map of relative exchange oscillation frequency per double-well, with the black rectangle indicating the region of the system used in **b**. Bottom, the decay of contrast in the region of interest (data points) is well described by a fitted Gaussian-type decay time of $\tau_{ex} = 33(2)$ ms (dashed line). **e**, Same as in **c** but for the $|\uparrow\downarrow,0\rangle$ state. The fitted Gaussian-type decay is $\tau_{ex} = 15(3)$ ms. All error bars are calculated as a 68% confidence interval, unless stated otherwise.

the Hamiltonian of equation (1); therefore, the theoretically expected final occupations for these initial states are shown in grey.

To extract the two-qubit gate fidelity, we apply up to 20 consecutive gates and fit an exponential decay to the population of the $|\uparrow,\downarrow\rangle$ state (Fig. 4c). We analyse the data on 64 lattice sites, with more than 175 experimental realizations per setting, and post-select on double-wells occupied by two particles. The offset extracted from the fit quantifies the state preparation and measurement (SPAM) error. The greyed-out data point, representing the measurement without any applied gate, shows artificially increased population as band excitations from the state preparation are initially misidentified and relax after the first applied gate (Methods). These SPAM errors also affect the truth table measurement. The fitting function accurately captures the remaining data points, yielding an average gate fidelity of 99.75(6)% for entangling two adjacent neutral atoms in the entire array. Using more sophisticated randomized benchmarking protocols, similar fidelities have been reported in fermionic platforms based on quantum dots[53,54], for individual double quantum dot devices.

To evaluate the lifetime of the entangled state, we perform a spin-sensitive Ramsey measurement. Starting from the $|\uparrow,\downarrow\rangle$ state, we apply a $\sqrt{\text{SWAP}}$ gate to entangle the atoms. A strong magnetic field gradient of 40 G cm$^{-1}$ is then applied along the double-well potential, at a Feshbach field of 686.9 G. This lifts the degeneracy between the $|\uparrow,\downarrow\rangle$ and $|\downarrow,\uparrow\rangle$ states by 9 Hz, resulting in a phase rotation of the entangled state. On the two-particle Bloch sphere, this corresponds to a rotation around the $z$-axis (Fig. 4d, left), inducing singlet–triplet oscillations[55]. A second disentangling $\sqrt{\text{SWAP}}$ pulse subsequently enables state readout in the single-particle basis $\{|\uparrow\rangle, |\downarrow\rangle\}$ (Fig. 4d, middle). The oscillations show close to full contrast over 1 s of measurement time, with thermal effects in the experiment limiting longer durations. We estimate a coherence time of more than 10 s (Methods), which is four orders of magnitude longer than the duration of the single entangling gate. This long coherence time is ensured by the small differential magnetic dipole moment in the Paschen–Back regime.

Much of the untapped potential of fermions in optical superlattices lies outside the demonstrated and robust spin-qubit framework. Notably, controlling fermionic motional degrees of freedom using gates enables opportunities for the use of ultracold fermions in materials science and quantum chemistry. In this context, a common approach is to endow an uncorrelated Hartree–Fock wavefunction with correlations through coherent excitations[3,15]. These can be single excitations, driven by single-particle tunnelling, or double excitations, created by correlated particle motion.

Using concatenated gate sequences, we demonstrate the engineering of specific double excitations. We focus on the pair-exchange process PX($\Theta$), which describes the correlated tunnelling of an unbroken fermionic pair between two spatial orbitals,

$$\text{PX}(\Theta) = \begin{bmatrix} 1 & 0 & 0 & 0 & & & \\ 0 & 1 & 0 & 0 & & \mathbf{0} & \\ 0 & 0 & 1 & 0 & & & \\ 0 & 0 & 0 & 1 & & & \\ & & \mathbf{0} & & \cos(\Theta) & -\sin(\Theta) \\ & & & & \sin(\Theta) & \cos(\Theta) \end{bmatrix}. \quad (3)$$

The Fermi–Hubbard model naturally realizes coherent pair-tunnelling[36] as demonstrated by the oscillations in Fig. 3d, but this process is usually coupled to other dynamics, such as spin-exchange or single-particle tunnelling. We develop and demonstrate a composite gate that specifically isolates the pair-exchange gate from these other processes (Methods). The main component of the composite gate consists of interaction dynamics that generate a discrete gate within the lower-diagonal 4 × 4 block of the unitary operator in equation (2). The truth table of a quasi-adiabatic Blackman interaction gate with rotation angle $\theta = \pi/2$ is shown in Fig. 5a. We then combine the interaction gate with a charge-sensitive dynamical tilt $\delta$ of the double-well ($Z$-gate; see equation (1)). To reduce the sensitivity to spatial inhomogeneities and temporal fluctuations, we devise a new

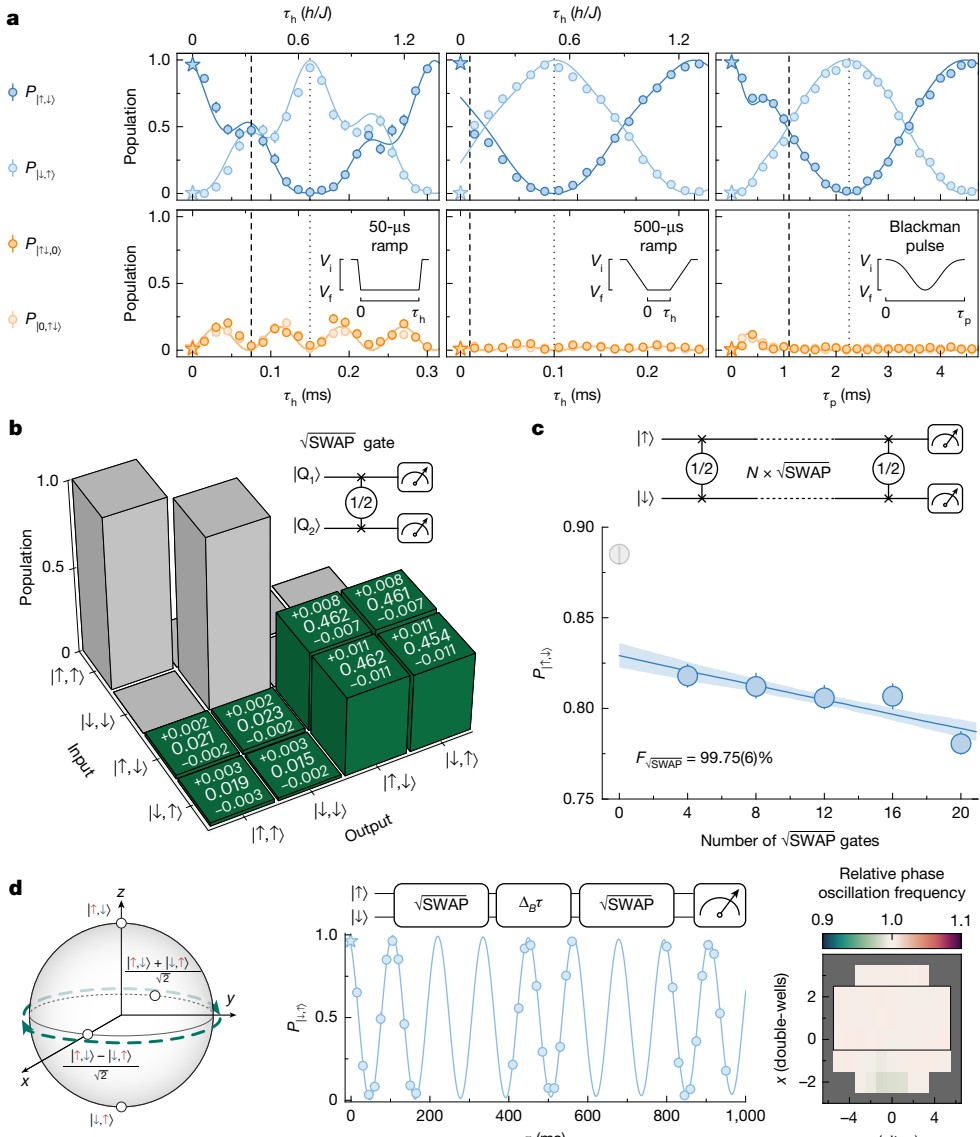

**Fig. 4 | High-fidelity SWAP$^\alpha$ gates. a**, Populations of states {$|\uparrow,\downarrow\rangle$, $|\downarrow,\uparrow\rangle$, $|\uparrow\downarrow,0\rangle$, $|0,\uparrow\downarrow\rangle$} in a double-well as a function of pulse duration for three different pulse ramp shapes: 50 µs linear (left), 500 µs linear (centre) and Blackman pulse (right) (see insets). In all cases, the system is initialized in $|\uparrow,\downarrow\rangle$ (indicated by star symbols). For linear pulses, we keep the ramp duration fixed and vary the hold time $\tau_h$, whereas for the Blackman pulse, we scan the total pulse duration $\tau_p$. The pulse durations corresponding to the $\sqrt{\text{SWAP}}$ (SWAP) gate are indicated by dashed (dotted) vertical lines. The solid lines are obtained from the simulation. A secondary $x$-axis shows the pulse duration in units of maximal spin-exchange frequency $h/J$. **b**, Measured truth table for the $\sqrt{\text{SWAP}}$ gate for input states $|\uparrow,\downarrow\rangle$

and $|\downarrow,\uparrow\rangle$. Input states $|\uparrow,\uparrow\rangle$ and $|\downarrow,\downarrow\rangle$ are not coupled by the natural Hamiltonian and the theory expectations are shown in grey. **c**, Return probability after several $\sqrt{\text{SWAP}}$ gates. The two-qubit gate fidelity of 99.75(6)% is extracted from an exponential decay fit of the state population (solid blue line). The uncertainty on fidelity is estimated using bootstrapping. **d**, Singlet–triplet oscillations are driven by a magnetic field gradient along the double-well potential axis. The data are well fitted by a damped sinusoidal with a lower bound on coherence time of 10 s, well beyond the measurement duration (Methods). A spatial map of measured Ramsey frequencies is shown on the right. All truth tables are shown without SPAM correction.

experimental protocol for these gate sequences (Fig. 5b and Methods). As phase errors scale with the depth of the long lattice, we generate the potential tilt $\delta$ near the maximally staggered configuration, for which even a shallow long lattice produces a strong differential tilt, making small phase errors much less impactful compared with previous experimental approaches[16]. A deep long lattice potential is used only during the $U_{int}$ pulses to confine atoms within each double-well. The optimized $Z$-gate enables the observation of fast and coherent Ramsey oscillations (Fig. 5c) with a frequency of tens of kHz. Residual spatial variation in Ramsey frequencies across the system (Fig. 5d) can be used to map out the spatial inhomogeneity of $\delta$.

We implement the pair-exchange gate PX($\Theta$) by interleaving $\theta = \pi/2$ and $\theta = 3\pi/2$ interaction pulses with a $Z$-gate. The states $|\uparrow,\downarrow\rangle$ and $|\downarrow,\uparrow\rangle$

(light and dark blue circles in Fig. 5e, top) are not affected by the tilt $\delta$ and perform a full $\theta = 2\pi$ rotation such that the population remains constant. On the other hand, for initial states $|\uparrow\downarrow,0\rangle$ and $|0,\uparrow\downarrow\rangle$, the final state depends on the double-well tilt $\delta$ and we observe an oscillation with a frequency of $2\delta$ (Fig. 5e, bottom). The reduced contrast of doublon oscillations originates from direct spin-exchange in the extended Fermi–Hubbard model[36,38], which slightly modifies the interaction gate in equation (2). We illustrate the effect of the sequence by the truth table of a PX($\Theta = 2\delta\tau = 7\pi/2$) gate shown in Fig. 5f. Unlike the native interaction gate (Fig. 5a), this gate selectively performs a $\sqrt{\text{SWAP}}$ on the charge sector while leaving the spin sector unaffected. This proof-of-principle realization marks an important step towards using fermions in optical superlattices for fermionic quantum computing,

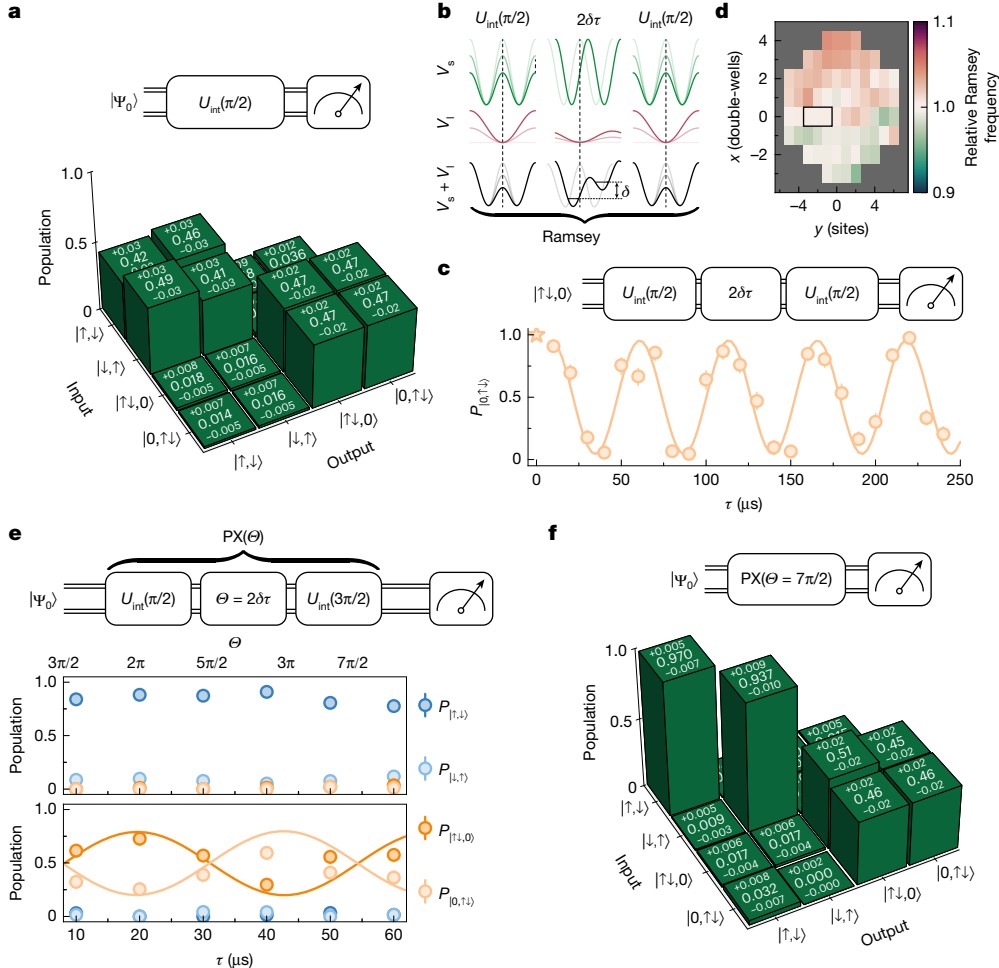

**Fig. 5 | Pair-exchange composite gate sequence. a**, Experimental truth table for the lower-diagonal 4 × 4 block of the interaction matrix $U_{int}$ with $\theta = \pi/2$. **b**, Superlattice potentials (short lattice in green, long lattice in red and total potential in black) used to implement the charge-sensitive Ramsey sequence composed of two $U_{int}$ and one $Z$-gate tilt. **c**, Ramsey oscillation in the population of $|\uparrow\downarrow,0\rangle$ for the sequence in **b**. **d**, Map of relative Ramsey frequencies per double-well, with the black rectangle indicating the region of the system used

in **c**, with an extracted frequency of 19.2(2) kHz. **e**, Applying a circuit of interaction and tilt pulses for the initial states $|\Psi_0\rangle = |\uparrow,\downarrow\rangle$ (top) and $|\Psi_0\rangle = |\uparrow\downarrow,0\rangle$ (bottom). This generates an effective pulse that only performs a coherent pair-tunnelling $|\uparrow\downarrow,0\rangle \leftrightarrow |0,\uparrow\downarrow\rangle$ without changing the spin states $|\uparrow,\downarrow\rangle$ and $|\downarrow,\uparrow\rangle$. The solid lines in **c** and **e** correspond to the simulated data. **f**, Experimental truth table of the pair-exchange gate PX($\Theta = 7\pi/2$).

for instance, for the simulation of non-native Hamiltonians[19] or variational methods in quantum chemistry[3].

The high-fidelity collisional entangling gates and the intrinsically long coherence times demonstrated here establish optical lattices as a compelling route towards scalable quantum computing. Reconfiguring the superlattice dimerization extends the platform beyond isolated dimers, enabling, for example, the generation of large-scale entanglement[39]. Collisional gate performance can be further improved through optical potential flattening[40–42] and optimal control techniques[28,44]. We can readily anticipate sub-10-μs entangling gates acting on systems as large as 10,000 lattice sites (Methods). Single-qubit gates and local control, as already realized in advanced Rydberg-based neutral-atom platforms[28–30,56], can be implemented through Raman transitions driven by tightly focused ultraviolet addressing beams, enabling randomized benchmarking[57] and full programmability.

Motivated by these prospects, we predict the following development of fermionic quantum simulators over the coming years: short circuits of the composite gates presented here will add digital state initialization and readout capabilities to analogue quantum simulators. The new observables and state preparation schemes enabled by this hybrid approach with global control will greatly enhance the utility of fermionic quantum simulators[17–19]. With further technical

progress, particularly for local control of motional quantum gates, purely digital fermionic schemes for universal computation[2] or variational methods for chemistry[3] will become feasible, probably based on two-dimensional superlattice architectures[39]. In the longer term, error-corrected fermionic circuits may lead to large-scale digital fermionic quantum simulation. Concrete architectures for fermionic error correction have been put forward[58,59] and we anticipate further rapid technical and conceptual developments in this active field.

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

## Methods

### Experimental platform

In our experiment, we prepare a degenerate Fermi gas of $^6$Li atoms in a balanced mixture of the two lowest hyperfine states, which represent our two spin states. The atomic cloud is loaded into a single plane of a vertical lattice following our previous work[41,51], with radial confinement provided by a blue-detuned box potential projected using a digital micromirror device (DMD)[40,41].

From there, the atoms are loaded into a two-dimensional square optical lattice in the $x$–$y$ plane with lattice constants $a_{x,\text{long}} = 2.28(2)$ μm and $a_y = 1.11(1)$ μm. A DMD pattern is chosen such that a flat central region of approximately 145 sites is surrounded by a low-density reservoir[60]. The chemical potential of the reservoir, tuned by the light intensity of the DMD, controls the particle density $\langle \hat{n} \rangle$ at the centre. We realize a state with an average of nearly two particles per lattice site (close to a band insulator) at lattice depths of $V_x^{\text{long}} = 9.0\, E_r^{\text{long}}$ and $V_y = 9.3\, E_r^{\text{short}}$. Dynamics are frozen by ramping the lattice depths to $V_x^{\text{long}} = 35.5\, E_r^{\text{long}}$ and $V_y = 45.0\, E_r^{\text{short}}$, leaving isolated single-wells with mainly two particles per site. Subsequently, we ramp up a second, short-spaced lattice along $x$ ($a_{x,\text{short}} = a_{x,\text{long}}/2$) over 25 ms, resulting in isolated, doubly occupied double-wells with total spin $S_z = 0$. In this experiment, the short lattices are generated by laser beams at blue-detuned 532-nm light incident at an angle of about 27°. The long lattice along the $x$-direction follows the same beam path, except that it is generated with red-detuned 1,064-nm light[14].

In all figures, data points were collected in a randomized sequence to prevent systematic bias.

### State preparation fidelity

The probability of realizing the desired state in the region of interest is approximately constant within a given dataset and mostly depends on the relative phase drift between the long and short lattices, as well as on the chosen atomic density. It is set largely by the fidelity of preparing an average occupancy close to two atoms of opposite spin per initial lattice site, which ranges from 60% to 85%. Deviations from the target state fall into two categories: (1) empty or singly occupied double-wells, which we remove by post-selection, and (2) double-wells containing three or more atoms, typically with population in higher lattice bands. Because these high-occupancy events can be mistaken for gate errors, we deliberately work at slightly lower atomic densities to suppress them, retaining between 45% and 65% of double-wells in analysis. Recent demonstrations of low-entropy band insulators in optical lattices suggest that considerably higher state preparation fidelity is attainable[7,13,61]. We note that the state preparation step does not affect the intrinsic performance of the individual gate operations.

### Lattice depth calibration

Lattice depth calibration is performed by measuring single-particle oscillations in a double-well, from which we extract the calibration factor by fitting the observed tunnelling rates to theoretical predictions across a range of lattice depths. An initial state consisting of a single particle in a double-well is prepared by adjusting the atom density and tilting the double-well potentials during loading, similar to our previous work[14]. We then remove the potential offset $\delta$, resulting in a symmetric double-well configuration at lattice depths of $V_x^{\text{long}} = 36.5\, E_r^{\text{long}}$ and $(V_x^{\text{short}}, V_y) = (56, 43)\, E_r^{\text{short}}$. Quenching the short $x$ lattice depth to a lower value initiates coherent oscillations of the population between the two sites in the double-well. In our analysis, we post-select double-wells containing exactly one atom.

In Extended Data Fig. 1a, we show an example calibration plot in which the calculated calibration curve aligns with the measured tunnelling frequencies with residuals less than 1.5% of $V_x^{\text{short}}$. The tunnelling frequency of intra-double-well oscillations $f_t = 2t/h$ is extracted by fitting a resonant two-level oscillation $[1 + \cos(2\pi f_t \times \tau_h)]/2$ to the population of one of the wells, which is then compared with the frequency expected from a band calculation (see our previous work[14] for more details).

To cross-check the lattice depth calibration, we measure spin-exchange oscillation in the $U/t \gg 1$ regime ($J \approx 4t^2/U$), in which virtual doublon-hole excitations are strongly suppressed (Extended Data Fig. 1b). We compare the frequency extracted from the fit to the oscillations (Extended Data Fig. 1b, upper row) with the calculated calibration curve (solid blue line) and find excellent agreement, consistent with the single-particle tunnelling calibration. The initial lattice depths in this case are $(V_x^{\text{short}}, V_y) = (56, 45)\, E_r^{\text{short}}$, $V_x^{\text{long}} = 39.5\, E_r^{\text{long}}$ and the Feshbach magnetic field is set to 688.2 G to control the on-site interaction strength $U$ through a Feshbach resonance. The long lattice depth $V_x^{\text{long}}$ is independently calibrated using lattice modulation spectroscopy through band-excitation energies to an accuracy of 5%.

### Experimental protocol

The spin-exchange process is initialized from the state $|\uparrow, \downarrow\rangle$ (Fig. 3) by linearly lowering the intra-double-well barrier from $54\, E_r^{\text{short}}$ ($t \approx 0$) to $5.54\, E_r^{\text{short}}$ ($t = h \times 2.9(1)$ kHz) in 500 μs, at on-site repulsive interactions $U = h \times 6.7(1)$ kHz corresponding to a ratio $U/t \approx 4/\sqrt{3}$. After a variable holding time $\tau_h$, the intra-double-well barrier is ramped back to its initial value in 500 μs. Coherent pair-tunnelling dynamics are induced under identical conditions and at the same ratio $U/t$, starting from the initial state $|\uparrow\downarrow, 0\rangle$.

The oscillation frequency and coherence shown in Fig. 3c,e are obtained by fitting the data patches in Fig. 3b,d individually with:

$$g(\tau_h) = \frac{1}{2}[1 + A\cos(2\pi f_J(\tau_h - \tau_0))]. \tag{4}$$

Here $A$ is the contrast, $f_J = J/h$ is the frequency of exchange oscillations and $\tau_0$ is the phase offset. The decay of contrast $A$ (shown in Fig. 3c,e) is in both cases compatible with a Gaussian decay $\propto e^{-(\tau_h/\tau_{\text{ex}})^2}$ that originates from a spatial average over several sites with inhomogeneous oscillation frequencies[62] (see the section 'Effect of spatial averaging on collisional gates').

The data in Fig. 4b,c use the quasi-adiabatic approach with Blackman pulses. The total pulse duration for the $\sqrt{\text{SWAP}}$ gate is tuned to 1.125 ms in Fig. 4b and 1.29 ms in Fig. 4c. The data in Fig. 4 are post-selected on having two-particles in a double-well and is not SPAM corrected. Experimental parameters are given in Extended Data Table 1.

### Fermi–Hubbard double-well simulation

To accurately describe the continuous exchange dynamics (Fig. 3b,d), we simulate the Fermi–Hubbard Hamiltonian (equation (1)) by exact diagonalization for a double-well with two particles of opposite spin with the QuSpin library[63]. The calculation of the Hubbard parameters $t$ and $U$ from the depths of the optical lattices and the phase of the superlattice is explained in the supplemental material of ref. 14.

The two-particle exchange dynamics are well reproduced by a simulation based on the experimental parameters in Extended Data Table 1. Three empirical modifications are added to the bare simulation to fit the data. First, we fine-tune the depth of the long lattice by 0.3% (5%) for the spin-exchange (coherent pair-tunnelling) oscillations, relative to the value expected from lattice-shaking experiments. Second, we observe a small chirp in the exchange frequency during the 30 ms oscillation time, which we attribute to a small gradual change of the lattice depth owing to technical heating. To account for this effect, we apply a linear correction to $V^{\text{short}}$ when calculating $U$ and $t$:

$$V^{\text{short}}(\tau_h) = V_0^{\text{short}} + \Delta V^{\text{short}} \tau_h \tag{5}$$

The slope $\Delta V^{\text{short}}$ was found to be $4(1) \times 10^{-3} E_r^{\text{short}} \text{ s}^{-1}$ for both the spin-exchange and the coherent pair-tunnelling dynamics. Finally, to account for dephasing effects, the simulation results were multiplied by a Gaussian envelope, with parameters extracted from the fits shown in Fig. 3c,e. Apart from these three adjustments, no free parameters were needed. Notably, key features such as the initial phase of the oscillations and deviations from pure sinusoidal oscillations arise intrinsically from the simulation of the double-well system.

We performed similar simulations for the different lattice ramps (Fig. 4a). In this case, the only free fitting parameter is the long lattice depth $V^{\text{long}}$, adjusted by 0.3% in all three cases. Owing to the short duration of the pulses used in this experiment, thermal drifts and the associated frequency chirp can be safely neglected and were therefore not included in the simulation.

To reproduce the Ramsey oscillations of Fig. 5c in simulation, we increase the experimental long lattice depth by 12% to calibrate the $U_{\text{int}}(\pi/2)$ pulses. Because the idealized simulation does not capture all residual inhomogeneities, we also reduce the simulated contrast by 8.7%, a value extracted from a sinusoidal fit to the data. The experimental parameters used for the pair-exchange gates in Fig. 5e are identical to those of Fig. 5c, except for the short-lattice depth during the $3\pi/2$ pulse. This depth is reduced to $3.3 E_r^{\text{short}}$, optimized so that the $|\uparrow,\downarrow\rangle$ and $|\downarrow,\uparrow\rangle$ initial states undergo the desired $3\pi/2$ rotation. With only this modification compared with Fig. 5c, the resulting simulation reproduces the experimental data of Fig. 5e with good agreement. This comparison confirms that the reduced contrast is mainly caused by direct-exchange processes, which become prominent at the very low lattice depths used there and slightly modify the effective exchange coupling $J$ in the spin and charge sectors. At higher lattice depths, in which direct exchange is negligible, these effects are suppressed, and uniformly high performance for all initial states is achievable.

## Effect of spatial averaging on collisional gates

The decay of the global spin-exchange contrast (Fig. 3c,e) arises from inhomogeneous local oscillation frequencies, which lead to a Gaussian envelope when averaging over several sites[62]. This behaviour is further supported by comparing the experimental data with simulations that incorporate site-resolved distributions of spin-exchange frequencies. To capture the spatial inhomogeneity, the relative spin-exchange frequency map from Fig. 3c is fitted with a two-dimensional Gaussian profile (Extended Data Fig. 2a). Averaging over this fitted spatial distribution yields the contrast decay shown by the black curve shown in Extended Data Fig. 2b. The grey error band represents the range of simulated outcomes obtained by shifting the centre position $(x_0, y_0)$ of the two-dimensional Gaussian fit within its 68% confidence interval. This result reproduces both the Gaussian form and the correct order of magnitude of the decay of contrast, confirming its consistency with inhomogeneous dephasing. Further sources of dephasing such as lattice disorder or temporal fluctuations are not included in the model and may further reduce contrast.

## Two-qubit fidelity estimate

The fidelity $F_{\sqrt{\text{SWAP}}}$ of the entangling gate is estimated from an exponential decay fit $P(N_p) = p_0 (F_{\sqrt{\text{SWAP}}})^{N_p}$ (Fig. 4c), in which $p_0$ is the initial state population and $N_p$ is the number of applied pulses.

With our fully spin-resolved and charge-resolved imaging, the two-qubit gates errors depend on states kept in the analysis, that is, the chosen qubit basis. In a pure spin quantum computer, all measured states involving doublons or holes can be trivially ignored, whereas in a full fermionic quantum computer, all states are physically relevant and contribute to the error of the gate. Extended Data Fig. 3a shows how the fidelity estimation depends on this choice. In the most general case for two-particle states, we post-select on having two particles in one double-well potential (Extended Data

Fig. 3a, light-blue circles and Fig. 4c). For a spin-qubit basis, the unphysical states are $|\uparrow\downarrow,0\rangle$ and $|0,\uparrow\downarrow\rangle$, but on the other hand, states $|\uparrow,\uparrow\rangle$ and $|\downarrow,\downarrow\rangle$ are not part of the $S_z = 0$ basis. Black circles correspond to post-selection of only $|\uparrow,\downarrow\rangle$ and $|\downarrow,\uparrow\rangle$ states. Extracted fidelities are shown in the legend and largely remain unaffected by post-selection.

The jump in the $|\uparrow,\downarrow\rangle$ population after applying the first entangling pulse can be explained by a state preparation error that is not captured by the post-selection. During the initial preparation, which should lead to two particles with opposite spins per site, it can happen for two atoms with identical spin states to occupy the same lattice site, residing in different motional bands. Following spin-dependent splitting, which is used for initial-state preparation, such configurations (for example, $|\uparrow,\uparrow\rangle$ or $|\downarrow,\downarrow\rangle$) are distributed in the excited band (one well) and the ground band (the other well). After vertical spin-splitting, which is used for the final detection, these atoms are displaced in opposite directions as a result of their band-dependent motion, making them indistinguishable from the target state $|\uparrow,\downarrow\rangle$. If a gate pulse is applied before vertical spin splitting, the atoms in higher bands tunnel out of the double-well and throughout the system and are wrongly detected as one of the unwanted two-particle states, which is removed by post-selection. This is clearly visible in Extended Data Fig. 4, which shows fractions of particles in each of the six two-particle states for the initial state $|\uparrow,\downarrow\rangle$: the population in the initial state decreases after applying the first pulse, whereas numbers in almost all other states increase at this step. The data point without any gates is thus omitted in the determination of the gate fidelity.

As shown in Extended Data Fig. 2, the measured fidelity is limited by the homogeneity of the system and thus depends on the system size. Scaling the 64-qubit system to 128 lattice sites results in a slight decrease in average fidelity to 99.3% (Extended Data Fig. 3b). In future experiments, larger system sizes and higher fidelities could be achieved by using larger lattice beams and flattening the potential using a DMD[40,41].

Throughout this work, the limited maximum $y$-lattice depth has been among the leading sources of gate infidelity because residual inter-well tunnelling can lead to gate errors or misidentification of the final states. In Extended Data Fig. 3c, we show the dependence of mean fidelity of the central 64 lattice sites for different maximal lattice depths. We find that the freezing lattice depth of $43 E_r^{\text{short}}$ is still on the rising slope of fidelity. Increasing the lattice depth provides a direct route to further improve fidelities and reduce particle losses.

## Dephasing protection of spin qubits

The dephasing protection of spin qubits originates from their low sensitivity to magnetic field gradients. At a Feshbach field of 688.0 G, the energetically lowest two $^6$Li spin states exhibit a differential magnetic moment of $\Delta\mu_{\uparrow\downarrow} \approx 5 \text{ kHz G}^{-1}$. For dephasing to occur, an energy difference between the product states $|\uparrow,\downarrow\rangle$ and $|\downarrow,\uparrow\rangle$ is needed, which scales as $\Delta E \propto \Delta\mu_{\uparrow\downarrow}\Delta_B$, in which $\Delta_B$ is a magnetic field gradient.

One way to test for unwanted phase evolution is shown in Extended Data Fig. 5. After preparing the Bell state $(|\uparrow,\downarrow\rangle + i|\downarrow,\uparrow\rangle)/\sqrt{2}$, we freeze the dynamics for variable hold times and then apply a disentangling pulse that maps the atoms onto $P_{|\downarrow,\uparrow\rangle}$. Fitting an exponential decay yields a decoherence timescale of 10(1) s. This timescale is limited by dephasing owing to residual magnetic field gradients, hence it serves as a lower bound on the actual coherence of the Bell state. This lower bound on the coherence of the system exceeds the 1.3 ms required for a single entangling pulse by four orders of magnitude, meaning that spin-qubit decoherence has negligible contributions to collisional gate fidelity.

In a second Ramsey experiment, we measure coherence by means of singlet–triplet oscillations in a magnetic field gradient, as shown in Fig. 4d. We observe oscillations of the population at a frequency of 8.72(5) Hz that are compatible with the expected $\Delta E = h \times \Delta\mu_{\uparrow\downarrow}\Delta_B$ for

a magnetic gradient of $\Delta_B = 40.1(1)$ G cm$^{-1}$. During the measurement time of 1 s, we observe negligible decay of the oscillation contrast. To quantify coherence time, we fit the data with both exponential and Gaussian decay models, yielding decoherence times of 125 s (28 s), with 68% confidence intervals from 25 s (5 s) to infinity. We use the profile likelihood method from the lmfit library[64] to estimate these confidence intervals. On the basis of these measurements, we can conclude a conservative lower bound of 10 s on the coherence of the spin Bell state.

### Sequence design and control parameters for interaction and pair-exchange gate PX($\Theta$)

The interaction gate $U_{\mathrm{int}}(\pi/2)$ in Fig. 5a is realized by lowering $V_x^{\mathrm{short}}$ from 54.0 $E_r^{\mathrm{short}}$ to 7.87 $E_r^{\mathrm{short}}$ in 0.6 ms, with $V_x^{\mathrm{long}} = 35.0\,E_r^{\mathrm{long}}$. The lattice depth ramp is shaped as a quadratic pulse, which, similar to the Blackman pulse, helps mitigate doublon excitations and offers a robust, experimentally convenient pulse shape. The analysis is limited to three double-wells in which we post-select on having two particles in one double-well; the truth table is not SPAM corrected. Because part of the data are obscured in Fig. 5a, we also present the data in Extended Data Fig. 6a.

For composite pulse sequences such as the Ramsey sequence shown in Fig. 5c and the pair-exchange gate illustrated in Fig. 5e,f, precise control of the relative phase $\theta$ between the states $|\uparrow\downarrow,0\rangle$ and $|0,\uparrow\downarrow\rangle$ is essential. This relative phase is directly linked to the bias $\delta$, which scales as $\delta \propto V_x^{\mathrm{long}}\sin(\varphi_{\mathrm{ls}})$. In a standard approach, in which the long lattice depth is held constant throughout the gate sequence, fluctuations or spatial gradients in the relative phase $\varphi_{\mathrm{ls}}$ between the long and short lattice potentials limit our performance. To mitigate this, we design an improved pulse sequence, shown in Extended Data Fig. 7, that is more robust against such unwanted fluctuations: to avoid that the phase $\theta$ accumulates outside the gate time, the long lattice is here off, as $\delta$ scales with the long lattice depth $V_x^{\mathrm{long}}$. Using a similar protocol to that in Fig. 5c, for the interaction gate (green rectangles), we ramp down the short lattice depth to induce intra-double-well tunnelling and ramp up the long lattice depth to confine the atoms in the double-wells. For the charge-sensitive $Z$-gate tilt (blue rectangle), we use a shallow long lattice with a large lattice phase $\varphi_{\mathrm{ls}}$. Because the error in $\varphi_{\mathrm{ls}}$ is absolute, using a large phase suppresses the error. The optimal choice is $\varphi_{\mathrm{ls}} = \pi/2$, in which the sensitivity to fluctuations is only quadratic. However, owing to technical constraints, the experiment was conducted at $\varphi_{\mathrm{ls}} = 0.3\,\pi$.

### Outlook and prospects for the experimental platform

The optical superlattice platform offers substantial scope for further advancement of system size[13] and gate performance discussed in this work. In terms of scalability and gate speed, combining light fermionic $^6$Li with three times shorter lattice spacings of 383.5 nm (already demonstrated in a quantum gas microscope[65]), much faster quantum gates and usable array sizes on the order of $10^4$ lattice sites become realistic. Band-structure calculations and a generalized spin-exchange expression (Supplementary Information) give conservative estimates of 135 kHz for the spin-exchange rate and 235 kHz for single-particle tunnelling (Extended Data Fig. 8), indicating that sub-10-μs gates are feasible. Optimal control pulse shaping could further shorten these times[43,44], whereas randomized benchmarking[57] will provide a comprehensive assessment of gate fidelity.

### Composition of the pair-exchange gate

Figure 5e illustrates the composite pair-exchange (PX) gate implemented in this work, which consists of a phase gate $U_Z(\theta)$ sandwiched between two interaction gates. The $Z$-phase pulse is applied by means of the bias $\delta$ of the double-well (see Hamiltonian in equation (1)) and results in:

$$U_Z(\Theta) = \begin{bmatrix} 1 & 0 & 0 & 0 & & \\ 0 & 1 & 0 & 0 & & \mathbf{0} \\ 0 & 0 & 1 & 0 & & \\ 0 & 0 & 0 & 1 & & \\ & \mathbf{0} & & e^{-i\Theta - i\zeta''} & 0 \\ & & & 0 & e^{i\Theta - i\zeta''} \end{bmatrix}.$$

Here $\Theta = 2\pi \times \delta\tau_{\mathrm{hz}}/h$ is the tilt phase arising from the energy offset $\delta$ and $\zeta'' = 2\pi \times U\tau_{\mathrm{hz}}/h$ is dependent on the on-site interaction.

The composite sequence *Int–Z–Int*, comprising two interaction gates and one $Z$-phase pulse, results in:

$$\mathrm{PX}(\Theta) = U_{\mathrm{int}}\!\left(\frac{3\pi}{2}\right) U_Z(\Theta) U_{\mathrm{int}}\!\left(\frac{\pi}{2}\right)$$

$$= \begin{bmatrix} 1 & 0 & 0 & 0 & & \\ 0 & 1 & 0 & 0 & & \mathbf{0} \\ 0 & 0 & 1 & 0 & & \\ 0 & 0 & 0 & 1 & & \\ & \mathbf{0} & & e^{-i\zeta'}\cos(\Theta) & -e^{-i\zeta'}\sin(\Theta) \\ & & & e^{-i\zeta'}\sin(\Theta) & e^{-i\zeta'}\cos(\Theta) \end{bmatrix}. \quad (6)$$

Here $\zeta' = 2\pi \times U\tau_{\mathrm{total}}/h$ is a $U$-dependent phase associated with the combined duration $\tau_{\mathrm{total}}$ of the three applied gates. This phase can be effectively cancelled by appending an appropriate waiting time at the end of the sequence. Because the tilt $\delta$ couples exclusively to the doublon-hole manifold and is invisible to the spin manifold, the protocol isolates pair-exchange from background spin-exchange.

Extended Data Fig. 9 shows the corresponding Bloch-sphere trajectories: the doublon-hole manifold (Extended Data Fig. 9b) traces a great-circle arc of angle $\Theta$, whereas the spin manifold (Extended Data Fig. 9a) executes a closed loop and returns to its origin. These trajectories verify that the composite sequence realizes the intended pair-exchange operation with high fidelity while leaving the spin sector untouched. Extended Data Fig. 6b shows the truth table for the diagram in Fig. 5f for PX($\Theta = 7\pi/2$).

### Note

After preparing the manuscript, we learned of a related realization of high-fidelity quantum gates for spin-exchange using fermionic atoms[66].

### Data availability

The datasets generated and analysed during the present study, as well as the code used for the analysis, are available from the corresponding author on request. Source data are provided with this paper.

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

**Acknowledgements** We thank B. Schiffer, J. Singh, J. A. P. Reuter and R. Zeier for the insightful discussions. This work was supported by the Max Planck Society (MPG), the Horizon Europe programme HORIZON-CL4-2022 QUANTUM-02-SGA (project 101113690, PASQuans2.1), the German Federal Ministry of Research, Technology and Space (BMFTR grant agreement

13N15890, FermiQP) and Germany's Excellence Strategy (EXC-2111-390814868). T.H. received financing from the European Research Council (ERC) under the European Union's Horizon Europe research and innovation programme (grant agreement no. 101165353 – ERC Starting Grant FOrbQ). P.M.P. acknowledges support from the European Union's Horizon 2020 research and innovation program (grant agreement no. 948240 – ERC Starting Grant UniRand). Open access funding provided by MPG.

**Author contributions** P.B. led the project and collected and analysed most of the data. S.W. and T.F. assisted with the data collection. T.F. performed the Fermi–Hubbard simulations. T.H. conceived the project. P.B. and T.F. wrote the manuscript. T.F., T.H. and I.B. supervised the study. P.B., T.H., S.W., J.O., M.B., D.T., T.C., P.M.P., I.B. and T.F. worked on the interpretation of data and contributed to the final manuscript.

**Funding** Open access funding provided by Max Planck Society.

**Competing interests** The authors declare no competing interests.

**Additional information**
**Correspondence and requests for materials** should be addressed to Titus Franz.

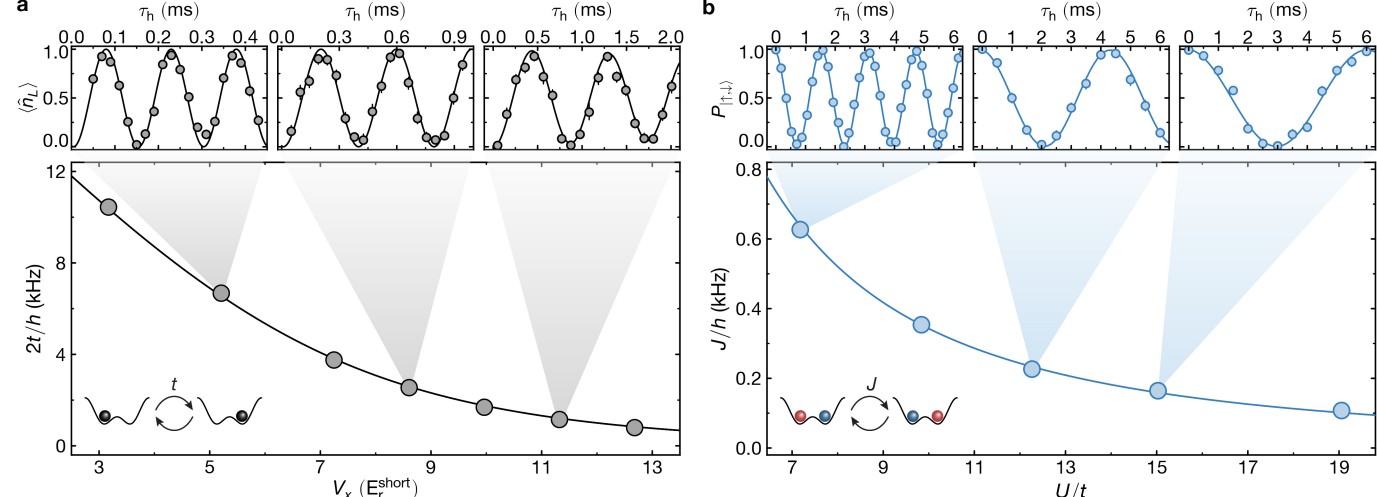

**Extended Data Fig. 1 | Calibration of single-particle tunnelling and spin-exchange oscillations. a**, Tunnelling frequency $2t/h$ of a single atom in a double-well for different $V_x^{short}$. The top row shows the time evolution of the population $\langle \hat{n}_L \rangle$ in the left site of the double-wells as a function of holding time $\tau_h$ following a quench to lower lattice depth at $\tau_h = 0$. The error bars denote one s.e.m. and are smaller than the marker when not visible. **b**, Extracted $J/h$ for different $x$-lattice depths compared with $J = 4t^2/U$ (solid line). The top panels show spin-exchange oscillations at different lattice depths.

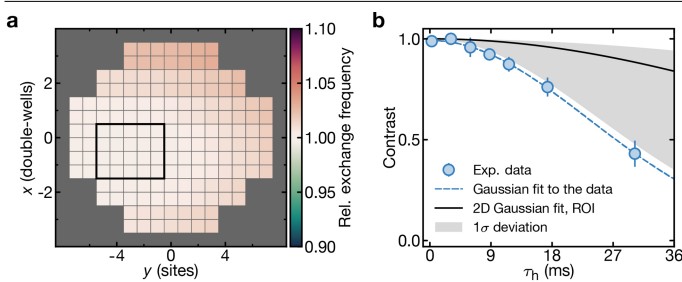

**Extended Data Fig. 2 | Ensemble dephasing owing to spatial frequency inhomogeneity. a**, Spatial map of the relative spin-exchange frequency, obtained by means of a two-dimensional Gaussian fit to the data shown in Fig. 3c. The black rectangle marks the region of interest (ROI) taken into account for the oscillation plots in Fig. 3b and the simulation. **b**, Contrast decay as a function of hold time $\tau_h$. Blue circles show experimental data, with a Gaussian fit (dashed blue line) indicating coherence decay. The solid black line shows the simulated contrast decay for the two-dimensional Gaussian obtained from the fit and the grey error band shows the range of results when shifting this fit by the 68% fitting uncertainties in the $(x_0, y_0)$ centre position.

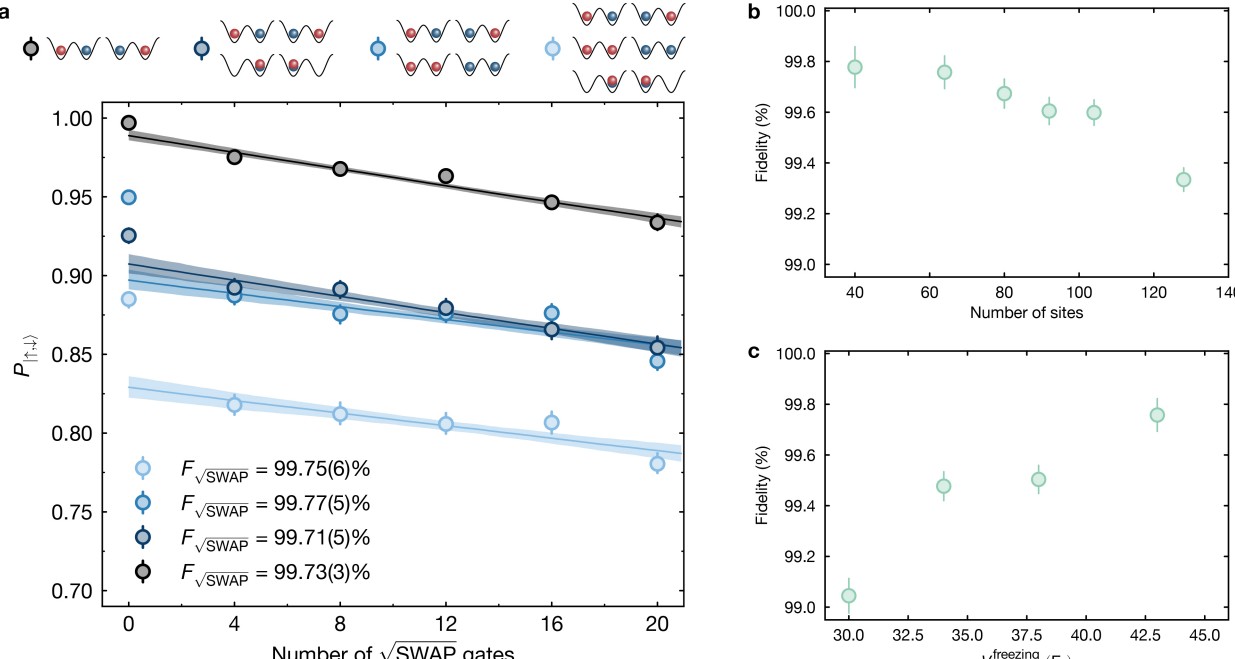

**Extended Data Fig. 3 | Two-qubit fidelity. a**, We extract fidelities for different states remaining after post-selection, as shown by the cartoons at the top. The solid lines of the respective colour show the exponential fits. **b**, Dependence of mean fidelity on the number of considered lattice sites. **c**, Mean fidelity of 64 lattice sites for different freezing lattice depths, showing that we are not yet in the saturation regime.

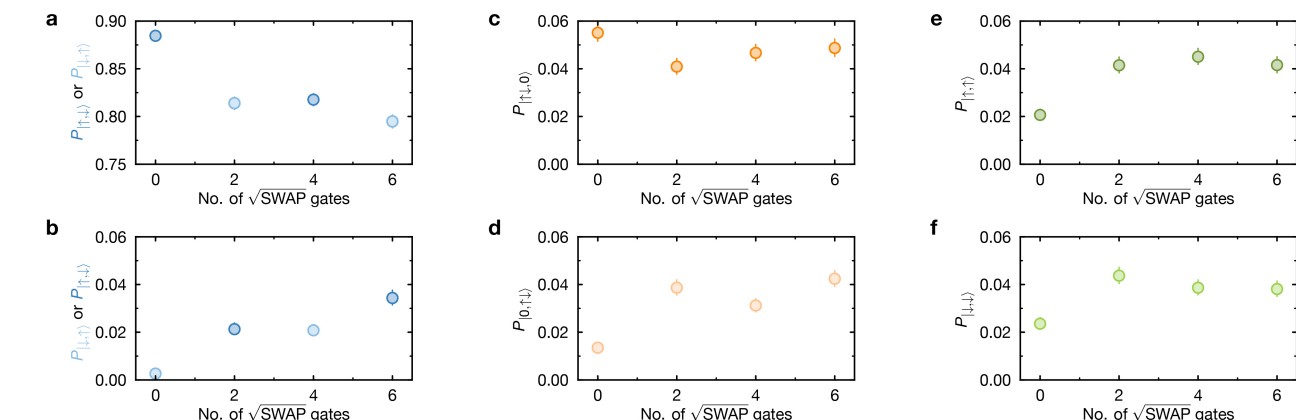

**Extended Data Fig. 4 | State preparation errors.** Measured fraction of two-particle states in $|\uparrow,\downarrow\rangle$ or $|\downarrow,\uparrow\rangle$ (**a**,**b**), $|\uparrow\downarrow,0\rangle$ (**c**), $|0,\uparrow\downarrow\rangle$ (**d**), $|\uparrow,\uparrow\rangle$ (**e**) and $|\downarrow,\downarrow\rangle$ (**f**) as a function of the number of $\sqrt{\text{SWAP}}$ pulses, for the initial state $|\uparrow,\downarrow\rangle$. Two $\sqrt{\text{SWAP}}$ gates perform a SWAP operation leading to alternating entries in **a** and **b**. Some state preparation errors such as band excitations are only detected after some time, resulting in the initial drop in **a**.

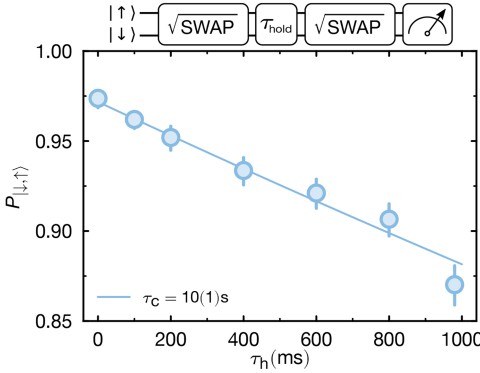

**Extended Data Fig. 5 | Coherence of Bell state.** Phase stability test of the entangled state using a variable hold time between two $\sqrt{\text{SWAP}}$ pulses. The first pulse prepares the state from $|\uparrow,\downarrow\rangle$ and the second maps the accumulated phase into the population for readout.

**a**

| Input | $|\uparrow,\downarrow\rangle$ | $|\downarrow,\uparrow\rangle$ | $|\uparrow\downarrow,0\rangle$ | $|0,\uparrow\downarrow\rangle$ |
|---|---|---|---|---|
| $|\uparrow,\downarrow\rangle$ | 0.42 +0.03 −0.03 | 0.46 +0.03 −0.03 | 0.018 +0.009 −0.006 | 0.036 +0.012 −0.009 |
| $|\downarrow,\uparrow\rangle$ | 0.49 +0.03 −0.03 | 0.41 +0.03 −0.03 | 0.031 +0.011 −0.008 | 0.028 +0.010 −0.008 |
| $|\uparrow\downarrow,0\rangle$ | 0.018 +0.008 −0.005 | 0.016 +0.007 −0.005 | 0.47 +0.02 −0.02 | 0.47 +0.02 −0.02 |
| $|0,\uparrow\downarrow\rangle$ | 0.014 +0.007 −0.005 | 0.016 +0.007 −0.004 | 0.47 +0.02 −0.02 | 0.47 +0.02 −0.02 |

Output

**b**

| Input | $|\uparrow,\downarrow\rangle$ | $|\downarrow,\uparrow\rangle$ | $|\uparrow\downarrow,0\rangle$ | $|0,\uparrow\downarrow\rangle$ |
|---|---|---|---|---|
| $|\uparrow,\downarrow\rangle$ | 0.97 +0.005 −0.007 | 0.000 +0.004 −0.000 | 0.007 +0.001 −0.000 | 0.000 +0.005 −0.004 |
| $|\downarrow,\uparrow\rangle$ | 0.009 +0.005 −0.003 | 0.937 +0.009 −0.010 | 0.009 +0.005 −0.003 | 0.000 +0.002 −0.000 |
| $|\uparrow\downarrow,0\rangle$ | 0.017 +0.006 −0.004 | 0.017 +0.006 −0.004 | 0.51 +0.02 −0.02 | 0.45 +0.02 −0.02 |
| $|0,\uparrow\downarrow\rangle$ | 0.032 +0.008 −0.007 | 0.000 +0.002 −0.000 | 0.46 +0.02 −0.02 | 0.46 +0.02 −0.02 |

Output

**Extended Data Fig. 6 | Truth table for four input states. a**, For the interaction gate, without SPAM error correction. **b**, For the pair-exchange gate (PX($\Theta$)), without SPAM error correction.

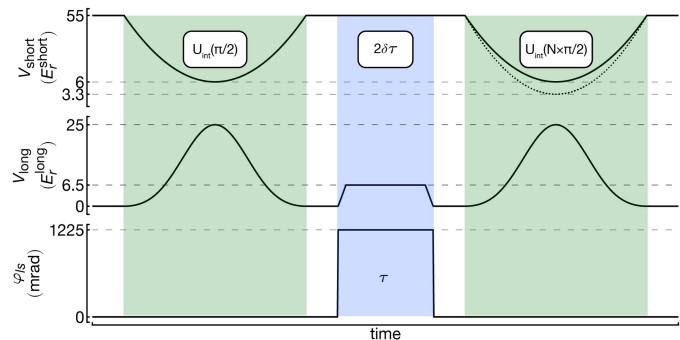

**Extended Data Fig. 7 | Time traces of pulses.** Experimental protocol for a charge-sensitive Ramsey (solid lines, $N = 1$) and PX($\Theta$) gate sequence (dashed lines, $N = 3$) that is robust to fluctuations and gradients of the phase ($\varphi_{ls}$) between long and short lattices. For the interaction gate (green rectangles), we ramp down the short lattice depth $V^{short}$ to induce inter-well tunnelling and ramp up the long lattice depth $V^{long}$ to confine the atoms in the double-wells. The total duration of $U_{int}$ gates is 1 ms. For the $Z$-gate tilt (blue rectangle), we use a shallow long lattice and large relative phase $\varphi_{ls}$ between long and short lattices to realize a controlled energy difference $\delta$ for varying duration $\tau$.

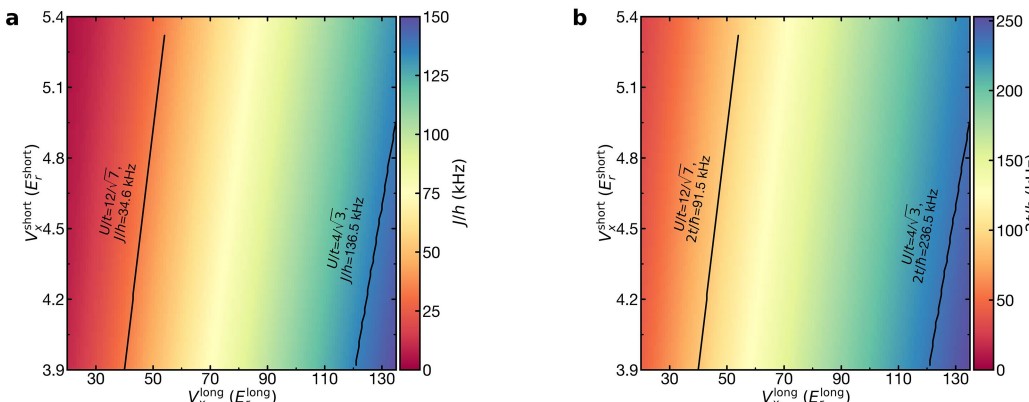

**Extended Data Fig. 8 | Expected parameters in an improved platform. a,b,** Expected spin-exchange frequency $J$ (**a**) and tunnelling frequency $t$ (**b**) for a lattice spacing of 383.5 nm, lattice depths of $V_y = 45\,E_r^{\mathrm{short}}$ and $V_z = 50\,E_r^z$ and Feshbach field of 767.2 G. Black lines represent magic $U/t$ ratios.

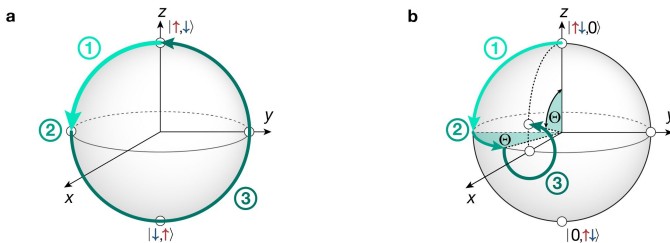

**Extended Data Fig. 9 | Illustration of the composite pulse for the pair-exchange gate (PX($\Theta$)).** Pulses 1 and 3 are interaction gates $U_{int}(\theta_i)$ with $\theta_1 = \pi/2$ and $\theta_3 = 3\pi/2$. Pulse 2 is a $Z$-gate $U_Z(\Theta)$. **a**, In the spin sector, pulse 2 has no effect on the states. Thus, pulses 1 and 3 together implement a full $2\pi$ rotation around the Bloch sphere. **b**, In the charge sector, pulse 1 rotates the initial state $|\uparrow\downarrow,0\rangle$ by $\pi/2$ around the $x$-axis of the Bloch sphere. In a general case, pulse 2 then rotates the state along the equator by an angle $\Theta$. The final pulse 3 acts as a rotation around the $x$-axis. For $\Theta = (2N+1)\pi/2$, this sequence realizes a pair-exchange gate.

**Extended Data Table 1 | Experimental parameters used in this work**

<table>
<tr><th colspan="3" align="center">Figure 3 of the main text</th></tr>
<tr><th>Parameter</th><th>Spin Qubit</th><th>DH Qubit</th></tr>
<tr><td>$V_x^{\text{short}}$ initial</td><td>$5.54\ E_r^{\text{short}}$</td><td>$6.00\ E_r^{\text{short}}$</td></tr>
<tr><td>$V_x^{\text{short}}$ freezing</td><td>$54.0\ E_r^{\text{short}}$</td><td>$54.0\ E_r^{\text{short}}$</td></tr>
<tr><td>$V_x^{\text{long}}$</td><td>$34.9\ E_r^{\text{long}}$</td><td>$36.8\ E_r^{\text{long}}$</td></tr>
<tr><td>$V_y^{\text{short}}$</td><td>$43\ E_r^{\text{short}}$</td><td>$43\ E_r^{\text{short}}$</td></tr>
<tr><td>Feshbach field</td><td>688.2 G</td><td>678.1 G</td></tr>
</table>

<table>
<tr><th colspan="4" align="center">Figure 4 of the main text</th></tr>
<tr><th>Ramp type</th><th>$V_x^{\text{short}}\ [E_r^{\text{short}}]$</th><th>$V_x^{\text{long}}\ [E_r^{\text{long}}]$</th><th>ROI [sites]</th></tr>
<tr><td>Fast linear<br>(Fig. 4a left)</td><td>5.6</td><td>34</td><td>128</td></tr>
<tr><td>Slow linear<br>(Fig. 4a middle)</td><td>5.6</td><td>36</td><td>128</td></tr>
<tr><td>Blackman<br>(Fig. 4a right; 4b)</td><td>6.5</td><td>38</td><td>128</td></tr>
<tr><td>Blackman<br>(Fig. 4c)</td><td>7.6</td><td>38</td><td>64</td></tr>
</table>

Continuous spin and doublon-hole (DH) exchange oscillations (left) and quantum gates with different ramp shapes (right). For the right panel, $V_x^{\text{short}}$ and $V_x^{\text{long}}$ are the minimal lattice depths of the short and long lattices, respectively. For all gates, the initial and final lattice depths are $V_x^{\text{short}} = 54\ E_r^{\text{short}}$ and $V_y = 43\ E_r^{\text{short}}$ and the Feshbach field is 689.9 G. For Fig. 4c the Feshbach field is 688.2 G.