## [Peer Review File · Nature]

High-fidelity collisional quantum gates with fermionic atoms

Corresponding Author: Professor Immanuel Bloch

Version 0:

Reviewer comments:

Referee #1

(Remarks to the Author)

This manuscript by Bojovic et al. present experiments realizing entangling quantum gates of pairs of fermionic atoms in effective double-well potentials. The approach is based on the native Hubbard model description, whose parameters are experimentally controlled with an optical superlattice potential and a magnetic field gradient, and analyzed using quantum gas microscopy. Focusing on the case of two atoms per double-well, the authors first demonstrate long-lived coherent oscillations of spin-exchange and pair-tunneling for appropriate initial states. They further characterize the underlying SWAP-type gate in detail, which is subsequently employed in a composite pair-exchange gate.

These experimental results are quite impressive. The claimed high fidelities and long coherence times show how fermionic atoms in optical lattices could become competitive also in a digital setting. Altogether this manuscript present a substantial development step towards more programmable quantum devices based on collisional gates with fermionic atoms. Before recommending publication, I would appreciate if the authors can resolve my technical questions (see further below), as well as some broader questions.

I am absolutely convinced about the usefulness of the developed gates for advanced state-preparation and read-out in analog-digital hybrids, but I remain more cautious about a fully digital mode because of the following more conceptual concerns.

1. In the abstract, the authors present the conservation of “symmetries like electron number” as a “distinct advantage”. I agree that this is useful in an analog setup. But it is unclear how much of an advantage remains in a digital setting. A more thorough discussion of advantages and disadvantages of the presented approach, in comparison to other platforms, could be useful. What do the authors see as the main competitive edge of their approach?

2. The question of error correction – which is the main focus of Rydberg-based neutral atom computing right now (notably Bluvstein et al., Nature 626, 58-65) – is only touched upon briefly in the last paragraph. According to Refs. 60 & 61, particle number conservation turns out to be a serious obstacle in building error-corrected fermionic processors. The present work points out that the basic gate can be employed for both spin and orbital degrees of freedom, but the requirements in an error-corrected setting are likely very different in those cases. I think many readers would like to see the authors’ opinion on this matter.

3. From the title I was expecting the advantage to come from the fermionic nature of the atoms, i.e. their fermionic statistics. But this is not mentioned throughout the whole text, although I appreciate that even a satisfactory theoretical answer of this (at least in an error-corrected setting) seems to be missing. On page 5, the authors briefly discuss the use of “controlling motional degrees of freedom [...] in material science and quantum chemistry”. Is this then meant in an analog (or hybrid) setting?

More detailed questions and comments:

a. I think there might some typos / sign errors in the derivations, if I am not misunderstanding some of the notation. In the SI, section 11, Eq. (S5), there is a minus sign in front of the matrix which appears to be in direct contradiction with Eq. (S4). Given the abundant potential for mistakes when dealing with fermions, I recommend making this part even more explicit. The

ordering ambiguity in the definition of the other states (e.g. $|\text{up down}, 0\rangle = + c_{\{L,\text{up}\}}^{\dagger} c_{\{L,\text{down}\}}^{\dagger} |0\rangle$?) is not spelled out. I quickly tried to reproduce Eq. (S10), but found an opposite overall sign as well as the opposite sign in the exponent in front of $U \tau_{\text{h}}$. Similarly, I found $\theta = + \pi/2$ (instead of $-\pi/2$) in line 1056 on page 16, as well as some different signs in Eq. (S3). Maybe I just made a mistake, but I would appreciate if the authors can double-check those.

b. In the context of a general-purpose fermionic processor, I was wondering about the phase of the tunneling t . For the physics of a double-well, this clearly does not matter because we can re-absorb it into the definition of the creation/annihilation operators. For a computer, this redefines the computational basis and would show up as a local relative phase between local occupied and unoccupied states. Have the authors considered such a phase?

c. Concerning the composite pair-exchange gate, Eq. (S3), the gate $U_z(\theta)$ is not defined explicitly. With an educated guess, I found Eq. (S3) up to a global θ -dependent phase. Please make this more explicit, ideally already in the main text. Is ζ' just equal 2ζ ? Additionally, I was wondering whether the decomposition only holds in a two-particle subspace or if it also works as part of a larger system (where different local particle numbers can appear).

d. In Figs. S11 & S12, the x-label should be $J \tau$, not τ / J ,

e. Throughout the main text, the authors state that data is post-selected on double wells that contain a specific number of particles. How much data is discarded? As also discussed in the SI, this removes some of the "unphysical" states of a spin qubit. However, such a post-selection is in general not possible in the middle of a circuit. For example, how can the authors be sure that a faulty gate does not lead to loss of atoms or rare tunneling into neighboring sites that do not belong to the double-well of interest? I think the presented fidelity should therefore be considered as a "best-case". It would also be interesting to see fidelities without post-selection.

f. I am not sure if I understand the effect of spatial inhomogeneity on the gate fidelities correctly. From the colormaps (e.g. Figs. 3c & S3a), it looks like the relative frequency error is at the percent level. From the description, it seems to me that this is a coherent effect that could in principle be fixed if one would adjust the hold times for each double-well individually. Assuming that it is just a mismatch of the hold time, a simple fidelity estimate from the oscillations would suggest $F \sim \cos(2\pi J t) = \cos(2\pi \epsilon)$ where ϵ is relative error on the timing $J t = 1$ arising from a slightly wrong $\delta J / J = \epsilon$. With $\epsilon = 0.01$ this gives $F = 99.8\%$. Do the authors agree with this? It further seems to me that the simulations of the experiment were of the same sort, i.e. including only coherent dynamics with systematic inhomogeneous calibration errors. Is this also correct? The fact that the data shows only Gaussian decay is compatible with only non-Markovian (quasi-static) noise, and makes me wonder about the magnitude of true incoherent (Markovian) noise. In this context, I would appreciate further discussion of fundamental errors sources. Can the authors provide a more complete error model of their device?

g. Further concerning the fidelity estimates. The analysis presented in Fig. 4c depends on the initial state. The established standard for estimating gate fidelities, however, is randomized benchmarking (as employed in Ref. 56). An alternative is gate set tomography (as employed in Ref. 57). Why have the authors not performed such a more rigorous analysis? Given the lack of such a benchmarking procedure, I find a direct comparison (of the number 99.75% estimated here) with those of Ref. 56 & 57 unjustified.

Referee #2

(Remarks to the Author)

"High-fidelity collisional quantum gates with fermionic atoms", by Bojovic et al, describes the use of a double-well potential with two fermionic neutral atoms to create quantum gates. Compared to systems that store quantum information in the spin of single atoms, and then mediate gates with Rydberg interactions, atoms in optical lattices offer collisionally mediated gates and thus intrinsic exchange-antisymmetry when working with fermions. Even in optical lattices, pioneering experimental work on quantum gates has used bosonic atoms. The significance of the current manuscript lies in the long-term prospect of quantum simulation of fermionic problems, such as finding the electron wave functions in molecules or solids. Here, the use of fermions as a building block in the quantum simulator is advantageous.

Towards this long-term direction, the specific advances reported are (1) the realization of two-atom gates with high fidelity and quality factor; (2) the demonstration of a composite gate that decouples doublon/hole dynamics from spin exchange dynamics; and (3) demonstration of a long-lived Bell state. These advances are significant, and will be considered a significant step forward in the lattice-based quantum simulation.

Before the article can be accepted, however, I encourage the authors to address several points.

A. Is the 'composite gate' demonstrated here without precedent, or can it be contextualized in prior work? The manuscript was not clear on this point.

B. What were the key challenges for realization of high fidelity gates? The authors discuss and quantify local inhomogeneities, but it would be useful for readers to know about other limiting factors that were improved in order to achieve 99.8% fidelity. Also, given the novelty of fermionic collisional gates, one wonders whether Pauli exclusion helps... or hinders.

C. I found the term "orbital" confusing, as a label of the doublon-hole degree of freedom. "Orbital" physics in optical lattices normally refers to higher-band physics, and is well established as a term. Even in the manuscript, the language shifts to "charge" (instead of "orbital") in later pages. I prefer this -- it is unambiguous.

D. One of the appeals of lattice-based QIP is its scalability. However, the authors demonstrate that fidelity depends sensitively on cloud inhomogeneities. With this perspective, the 99.8% fidelity achieved here is not as impressive as similar fidelities achieved in prior work, for much larger systems (since they were not site resolved). Comparing these two scenarios would be helpful for the reader to contextualize the work. Are inhomogeneities more severe in the author's apparatus, and if so, why?

E. The review of prior work (lines 50-65, where refs 5, 30-38 are discussed) is not clear. It would be helpful to give more specifics -- for instance, that all of those prior experimental works concern bosons.

F. Why were the specific magnetic fields (688G, 678G) chosen? Please clarify in the relevant Supp Mat section. Similarly, a gradient (40G/cm) is mentioned on line 277-- but this is not meaningful to most readers. Rather, giving the bias in terms of the energy scales of the problem (J or t) would improve clarity.

G. Figure 1a is not needed, nor more clear than the corresponding text without the figure parts.

H. I found the sentence spanning lines 328 to 333 unclear. ("Most importantly, the new protocol allows to implement the Z-gate ...") Is this connected to the following sentences? Please improve writing clarity here.

I. The "exponential fit" mentioned on line 789 is confusing. The data in Fig S4 appear to be fit with lines, based on their uncertainty bands. Is it more correct to say that a "linear fit" is made, which is interpreted assuming that it is the short-time limit of a simple exponential decay with no offset?

Referee #3

(Remarks to the Author)

The work of Bojovic and co-authors advances the pairwise control and entanglement in the spin and charge/motion sectors with 6Li atoms. The main motivation is recent proposals – for example Ref. [2], programmable gate-based approach that renders fermionic degrees of freedom in the hardware, or Ref. [11], directly implementing molecular simulations. The work presented here makes interesting progress on these topics, harnessing the fermionic atom that shows most promise for low-entropy systems 6Li, but does not fully convince me that the rate of advance, fidelity, and required control sequences mean these proposal will succeed at reaching the goals set out in those proposals.

In this portion of the discussion I will first discuss spin-exchange entanglement studies, and then pair tunneling and the pair exchange gate. At the end I discuss a number of other topics, such as the lack of individual control in this platform, which will be a large perturbation on the requirements of the platform.

I found the spin-exchange portion, which is the first topic presented, and also the highest fidelity, to be the least novel of topics presented in this work. To accomplished with 6Li is an advance that could help with integration, but the physics demonstrated is a repetition of an effect that has been seen again and again starting with the work of Anderlini et al (Ref. [36]). The effect is available for both bosons and fermions, with just a sign of the symmetry changed. In the context of the proposals above, it is also the most 'replaceable' in the sense that Rydberg gates could, in principle, be substituted (Ref. [2]). 96% fidelity was previously realized (Ref. 38), with local control in the verification, and this work increases to 99.8%, which is larger, but still not to the level of state of the art neutral atom gates. The coherence achieved of $J \tau_{\text{ex}} / \hbar = 110$ is dominated by inhomogeneities, and the fundamental reason for robustness was already clear in the early work of Ref. [36].

As the manuscript progresses into pair tunneling and pair exchange gate, the fundamental sensitivity to motion becomes (even more) important. This is the key challenge in harnessing the native fermionic nature in a variety of protocols. Consequently, the spatial dependence, dephasing, etc. are worse than the spin-exchange case, and represent the key benchmarks for the proposed protocols. For the pair-tunneling gate, I found the (significant) sensitivities and issues well described in the paragraph starting line 186.

For the pair-exchange gate, the authors state that this is a "proof of principle" result, and the fidelity is clearly challenging for the authors. While Fig. 5 provides some context for the overall quality of the gate via the green histograms, I would suggest the authors provide a more compact comparative description of the fidelity. And explain in detail how this fidelity compares to what is required for the proposed protocols.

As a general comment the phrase 'quantum chemistry' is used loosely throughout the paper in the context of a variety of protocols. The mapping between the demonstrations here and quantum chemistry requires a variety of logical steps, and the authors could be more specific in their description.

I now turn to the topic of local control of the motional gates, which the authors indicate is a clear next step in the conclusion. I found this at odds with a statement earlier in the paper that indicates fidelities can keep being pushed up via for example 'optical flattening techniques'. While the results here benefit from local measurements, local control is going to be a separate challenge. Single-site control will require a variety of new techniques and concepts that will make tweaks to the fidelities observed here irrelevant. A topic left to the supplement is the pair preparation fraction. This is 60-85% - this number and details of its origin, and impact on operations that do not involve post selection, I believe would best fully described in the

main text.

Fig. 5b would be clearer with the very small cartoon at the top better explained.

The authors indicate Eq. 2 “closely resembles the structure of a Molmer-Sorensen gate” – the authors should explain more clearly in what way this is key to the physics discussed in the paper.

Version 1:

Reviewer comments:

Referee #1

(Remarks to the Author)

I appreciate the authors' detailed response and resulting changes to the manuscript. All my concerns have been addressed, and I have no further questions or comments. I also read the response to the other referees' reports and find the authors' rebuttal satisfying.

Altogether, I recommend the revised manuscript for publication.

Referee #2

(Remarks to the Author)

The authors have addressed all concerns raised in my first round of review. I support publication without further changes.

Referee #3

(Remarks to the Author)

I thank the authors for the substantive changes made to the manuscript. In particular, the conclusion paragraph beginning “Motivated by these prospects...” now delineate the steps that exist between the experimental demonstrations in this work and digital fermionic computing with applications in quantum chemistry, although as per some of my original comments, the intermediate steps are multi-fold.

The responses to my queries and that of Referee B have clarified some of the relative importance and implications of the spin-exchange and pair-exchange gates. Although it remains that the spin-exchange gate, while improved quantitatively here, is in analogy to multiple prior demonstrations, both in ensemble and particle-resolved versions. The pair-exchange gate is now presented more clearly as a novel protocol with specific implications (eg circa line 470).

My main reservation remains in the role of fermionic nature in the experimental results and the associated manuscript discussion. Both Referee A and myself were surprised to find certain implications in the article and title, given, as the authors state “Our present gates do not yet rely on fermionic exchange statistics.” With this in mind, the authors should (1) specifically discuss this fact in the manuscript (2) continue to discuss bosonic realizations, eg for example leave in the discussion surrounding Refs. [45,61], in the introduction. The authors should also consider whether the title is clearly the right match. In the responses, the authors discuss positive aspects of both the fermionic nature of ^6Li , as well as the parameters achieved harnessing a low-mass atom.

High-fidelity collisional quantum gates with fermionic atoms

Authors: Bojović, Hilker, Wang *et al.*

INTRODUCTORY STATEMENT

We thank the editor and the referees for their careful reading of our manuscript and for their constructive and insightful comments. We have revised the main text and the Supplementary Information to address all points raised. The revised version includes an expanded discussion of the advantages and limitations of collisional gates, clarifications regarding sign conventions and matrix definitions, and additional explanations of the experimental methods and error sources. The overall structure and conclusions of the paper remain unchanged; however, the revised text should now provide a more precise and more comprehensive account of the scope, technical details, and context of our results.

LIST OF CHANGES TO THE MANUSCRIPT

- ◇ We modified the abstract and introduction to better reflect the practical advance offered by our approach in enabling future research.
- ◇ We added a discussion of the role of particle-number conservation and its implications for error correction, following the referees' suggestion.
- ◇ We clarified terminology by consistently using "charge" rather than "orbital" when referring to the doublon-hole degree of freedom.
- ◇ We revised the paragraph describing the Z-gate sequence to explain the use of a relative lattice phase $\varphi_{\text{ls}} \approx \pi/2$ and its robustness against phase fluctuations. To support this explanation, we updated Fig. 5b, moved the old figure to the Methods, and adjusted the captions accordingly.
- ◇ We removed several nonspecific mentions of quantum chemistry from the main text and made the connection between the pair-exchange gate and the relevant proposal more explicit.
- ◇ We made minor stylistic corrections and improved figure captions for clarity (Figs. 1a, 5).
- ◇ We replaced the previously fitted curves with simulations in Fig. 5c and Fig. 5e.
- ◇ We have substantially revised and expanded the conclusion to more clearly articulate the platform's competitive advantages while transparently addressing current limitations. The new text structures our vision into two distinct phases: near-term applications in hybrid analog-digital simulation and the long-term trajectory toward universal, error-corrected fermionic quantum computing.

LIST OF CHANGES TO THE SUPPLEMENTS

- ◇ Made the matrix derivations in Sec. 11 more explicit, corrected overall sign conventions, and added intermediate steps and definitions of state ordering to avoid ambiguity

- ◇ Added a clear definition of the Z-gate operator $U_Z(\theta)$ and its relation to ζ in Eq. (S3).
- ◇ We improved the description of state-preparation fidelity and clarified that it does not affect the final gate fidelity.
- ◇ Extended the discussion of experimental parameters to specify magnetic fields, gradients, and their energy-scale equivalents.
- ◇ Clarified the choice of Feshbach fields and the operational conditions for the pair-exchange sequence (Sec. 8).
- ◇ We added Fig. S8 from the main text to Sec. 8 to support the expanded explanation of the composite gate sequence. The new description explains this effect more clearly.
- ◇ We have expanded Sec. 9 to detail how the system size can be scaled to comparable or larger sizes than those demonstrated in prior studies.
- ◇ Minor textual improvements and cross-references for consistency between the main text and the SI.

For our detailed response to the questions raised by the external experts, please refer to the end of this document.

RESPONSE TO REPORT FROM THE FIRST REFEREE (A)

We thank the referee for their thoughtful and detailed report. We greatly appreciate their positive assessment of the experiments, especially their recognition of the long coherence times, the high fidelities, and the potential of fermionic atoms in optical lattices to play a role in a digital setting. We are glad that the referee considers our work a substantial step towards more programmable quantum devices based on collisional gates, and values the usefulness of the developed gates for advanced state preparation and readout in analog-digital hybrid approaches.

At the same time, we acknowledge the referee's broader concerns about the prospects for fully digital operation, as well as the specific technical questions they have raised. We are grateful for this opportunity to clarify these points. In the revised version, we provide additional explanations and clarifications in line with the referee's comments. Below, we reply to each point in detail.

These experimental results are quite impressive. The claimed high fidelities and long coherence times show how fermionic atoms in optical lattices could become competitive also in a digital setting. Altogether this manuscript present a substantial development step towards more programmable quantum devices based on collisional gates with fermionic atoms. Before recommending publication, I would appreciate if the authors can resolve my technical questions (see further below), as well as some broader questions.

We thank the referee for the positive assessment and address all technical and broader questions in detail below.

I am absolutely convinced about the usefulness of the developed gates for advanced state-preparation and read-out in analog-digital hybrids, but I remain more cautious about a fully digital mode because of the following more conceptual concerns: In the abstract, the authors present the conservation of "symmetries like electron number" as a "distinct advantage". I agree that this is useful in an analog setup. But it is unclear how much of an advantage remains in a digital setting. A more thorough discussion of advantages and disadvantages of the presented approach, in comparison to other platforms, could be useful. What do the authors see as the main competitive edge of their approach?

(A1) We appreciate the referee's comment and the opportunity to clarify this point. The advantage we refer to concerns the intrinsic conservation of fundamental symmetries in our system in comparison to qubit-based quantum computers. Because our platform employs real, massive fermionic atoms, the total particle number is strictly conserved, independent of gate fidelity or decoherence. Due to the SU(2) symmetry of the Hamiltonians for all gates, also the total magnetization (and total spin) remains conserved by construction. In contrast, in conventional qubit-based realizations, such as those based on spin-1/2 representations, a bit-flip error effectively corresponds to the creation or loss of a particle, and analogous processes can violate spin conservation. Using fermions directly thus naturally restricts the Hilbert space to the one for fermionic particles, thereby intrinsically preventing any non-physical

states. To make this distinction clearer, we have revised both the abstract and the main text to address the points raised by the referee.

The question of error correction – which is the main focus of Rydberg-based neutral atom computing right now (notably Bluvstein et al., Nature 626, 58-65) – is only touched upon briefly in the last paragraph. According to Refs. 60 & 61, particle number conservation turns out to be a serious obstacle in building error-corrected fermionic processors. The present work points out that the basic gate can be employed for both spin and orbital degrees of freedom, but the requirements in an error-corrected setting are likely very different in those cases. I think many readers would like to see the authors’ opinion on this matter.

(A2) We agree that error correction will ultimately be essential for scalable quantum digital computation with fermions. The referee correctly points out that particle-number conservation prohibits the straightforward implementation of error-correcting protocols for qubits. Recent theoretical proposals have begun to address this question, presenting the first proposals for error-corrected fermionic architectures, including both particle-number-non-conserving [60] and particle-number-conserving [61] approaches). Research in this area is evolving rapidly, and we now cite these works more prominently in the revised manuscript. The error correction schemes presented there could be adapted to work on our platform, although a complete experimental implementation will require further technical progress.

The field of hybrid, digital, and error-corrected fermionic quantum processors is evolving rapidly. To reflect this, we have included our perspective on likely future developments in this area within the outlook section. Our gates are compatible with the new proposals for error correction. In the near term, we aim to expand our approach toward a small-scale NISQ device, providing a testbed for further developments in error-mitigation and correction protocols. Moreover, we would like to emphasize that fermionic gates will be beneficial even without full error correction, particularly in hybrid analog-digital quantum simulations, where a small number of high-fidelity gates can already enable advanced state-preparation and readout protocols.

From the title I was expecting the advantage to come from the fermionic nature of the atoms, i.e. their fermionic statistics. But this is not mentioned throughout the whole text, although I appreciate that even a satisfactory theoretical answer of this (at least in an error-corrected setting) seems to be missing.

(A3) Our present gates do not yet rely on fermionic exchange statistics. The full advantage of using fermionic atoms will become apparent only once extended two-dimensional geometries and more complex motional circuits are realized. Such circuits will, in fact, build on our demonstrated two-qubit gates for fermions, which form essential ingredients.

In standard qubit-based systems, fermionic Hamiltonians must be mapped to spin degrees of freedom through schemes such as Jordan-Wigner (with linear overhead in N) or Bravyi–Kitaev (with gate depth and locality logarithmic in N , but a large constant prefactor), as well as more advanced methods exploiting conserved symmetries or parity sectors [28, 29, 4]. Notably, superconducting and trapped-ion architectures have recently reached the regime of computing controlled tunneling dynamics, which,

however, remains experimentally still very limited to very small systems and evolution times smaller than 0.8 tunneling times [38, 39], highlighting the challenge of simulating fermions for traditional quantum computing devices. In this sense, the use of fermionic atoms is a forward-looking choice: it enables a route toward digital fermionic quantum computation, complementing recent advances in spin-based simulators that encode fermions [8, 13].

On page 5, the authors briefly discuss the use of "controlling motional degrees of freedom [...] in material science and quantum chemistry". Is this then meant in an analog (or hybrid) setting?

(A4) This statement refers to a future digital (fermionic) quantum computer based on motional gates. Given the complexity of multi-band materials and molecules relevant for applications, we believe that programmable digital gates are the most promising approach for quantum simulation of these systems.

I think there might be some typos / sign errors in the derivations, if I am not misunderstanding some of the notation. In the SI, section 11, Eq. (S5), there is a minus sign in front of the matrix which appears to be in direct contradiction with Eq. (S4). Given the abundant potential for mistakes when dealing with fermions, I recommend making this part even more explicit. The ordering ambiguity in the definition of the other states (e.g. $|\uparrow\downarrow, \theta\rangle = \pm c_{L,\uparrow}^\dagger c_{L,\downarrow}^\dagger |\emptyset\rangle$?) is not spelled out. I quickly tried to reproduce Eq. (S10), but found an opposite overall sign as well as the opposite sign in the exponent in front of $U\tau/\hbar$. Similarly, I found $\theta = +\pi/2$ (instead of $-\pi/2$) in line 1056 on page 16, as well as some different signs in Eq. (S3). Maybe I just made a mistake, but I would appreciate if the authors can double-check those.

(A5) We thank the referee for their careful reading and for pointing out the potential inconsistencies in the sign of the results in the Supplemental Information (SI). The referee is indeed correct that, in the submitted version, sign inconsistencies and typos appeared between Eqs. (S3)-(S5) and the subsequent derivations.

We have now carefully rechecked all derivations and ensured that the sign conventions are consistent throughout both the main manuscript and the SI. In particular,

- ◊ The minus sign in front of the matrix in Eq. (S6) (previously Eq. (S5)) has been corrected for consistency with Eq. (S4).
- ◊ The operator ordering and phase conventions for the fermionic basis states (e.g., $|\uparrow\downarrow, 0\rangle = c_{L,\uparrow}^\dagger c_{L,\downarrow}^\dagger |\emptyset\rangle$) are now explicitly stated in Eq. (S5).
- ◊ The derivation of Eq. (S13) (previously Eq. (S10)) has been checked and rewritten in a clearer step-by-step form, ensuring consistency of all global and relative phases.
- ◊ The signs of the angles (e.g., $\theta = -\pi/2$) and of the exponents in $e^{-iU\tau/\hbar}$ have been verified and made consistent with the adopted convention for time evolution $U(t) = e^{-iHt/\hbar}$.

We have also expanded Section S11 of the SI by adding intermediate derivation steps, making the conventions and sign logic more explicit and thereby reducing any potential ambiguity for readers reconstructing the results.

In the context of a general-purpose fermionic processor, I was wondering about the phase of the tunneling t . For the physics of a double-well, this clearly does not matter because we can re-absorb it into the definition of the creation/annihilation operators. For a computer, this redefines the computational basis and would show up as a local relative phase between local occupied and unoccupied states. Have the authors considered such a phase?

(A6) We thank the referee for their insightful question. We understand the term "phase of tunneling" in two different ways, and we address both interpretations below.

From a microscopic perspective, the tunneling matrix element t originates from the overlap of neighboring Wannier functions. We agree with the referee's viewpoint that there is a gauge degree of freedom in the choice of phases for the Wannier functions, and that such phases become important in extended Hubbard models. For the separable lattice such as ours, the Wannier functions can always be chosen to be real (see the extensive treatment in [3]). The tunneling matrix element t is then given by the matrix element of the Hamiltonian operator between two neighboring Wannier states, $t = \left\langle w(x) \left| -\frac{\hbar^2 \nabla^2}{2m} + V(x) \right| w(x+a) \right\rangle$, which will also be real. Without an artificial magnetic field, we do not see how additional phases on t could arise.

In addition, there are dynamical phases accumulated during a tunneling operation in a digital-gate context, just as in other qubit rotations, e.g., a factor i after a $\pi/2$ pulse. The referee is correct in pointing out that, contrary to a spin-based quantum computer, where these phases are global and thus irrelevant, they do become relevant if the quantum processor is in a superposition between different particle numbers in a double well. One thus has to keep track of these phases and compensate for them, if necessary, with additional phase gates in the charge sector, i.e., potential offsets. We therefore do not view such phases as a fundamental limitation, but rather a technical detail that can be addressed with appropriately designed gate sequences [19].

Concerning the composite pair-exchange gate, Eq. (S3), the gate $U_z(\Theta)$ is not defined explicitly. With an educated guess, I found Eq. (S3) up to a global Θ -dependent phase. Please make this more explicit, ideally already in the main text. Is ζ' just equal 2ζ ? Additionally, I was wondering whether the decomposition only holds in a two-particle subspace or if it also works as part of a larger system (where different local particle numbers can appear).

(A7) We thank the referee for bringing this to our attention. As suggested, we have now explicitly defined $U_z(\Theta)$ in the Methods section for clarity and to maintain a compact main narrative. We have also detailed the origin of ζ' , which is determined by the interaction strength U and the total duration of the composite pair-exchange gate sequence.

The decomposition also applies within a larger system (see e.g. [19], reference [11] in the main text, Appendix E: "Universal Experimental Control Set", particularly Fig. S4a,b), where a universal control set enabling individual addressing of both the single- and two-particle sectors is derived. In principle, using composite gate sequences, even a single local control parameter—such as the potential offset—would suffice to render the universal control local. However, implementing this in practice requires additional

local and global control elements, as well as more complex composite pulse sequences. These advanced control capabilities are currently nearing operation in a new apparatus.

In Figs. S11 & S12, the x-label should be $J \tau$, not τ / J ,

(A8) We appreciate the referee's close reading of the manuscript and we have fixed the x-label.

Throughout the main text, the authors state that data is post-selected on double wells that contain a specific number of particles. How much data is discarded? As also discussed in the SI, this removes some of the "unphysical" states of a spin qubit. However, such a post-selection is in general not possible in the middle of a circuit. For example, how can the authors be sure that a faulty gate does not lead to loss of atoms or rare tunneling into neighboring sites that do not belong to the double-well of interest? I think the presented fidelity should therefore be considered as a "best-case". It would also be interesting to see fidelities without post-selection.

(A9) The referee is correct that our form of post-selection is not possible in an extended system, but needs to be replaced by erasure conversion or other error mitigation techniques. We use post-selection dominantly to remove imperfectly prepared initial states and show that it does not affect the gate performance. As shown in Fig. S4, applying different filters changes the inferred fidelity only at the few- 10^{-4} level. Fully unfiltered data (Fig. R1b) still yields a fidelity of 99.64(10)%.

In our post-selection (which we agree cannot be applied within a running circuit) we retain between 45% and 65% of the double-wells within the region of interest. To assess the influence of faulty gates, we routinely perform several control checks:

- ◊ **Atom loss.** For the data of Fig. 4c, we monitor the mean atom density per double-well as a function of the number of applied gates. The linear fit in Fig. R1a yields an atom loss rate of

Figure R1: **a**, Relative change of atom number for different holding times τ_h in Fig.S7. Data is plotted for the region of interest (blue points) and the full system size, including the reservoir (orange points). **b**, Inferred fidelity from the unfiltered data.

0.0008 ± 0.0010 atoms per gate, which is compatible with zero loss. Therefore, we conclude that atom loss does not play a significant role in the gate fidelity.

- ◇ **Tunneling to neighboring sites.** At a transverse lattice depth of $43E_r^{\text{short}}$, the coupling between adjacent sites is suppressed to just 0.5 Hz. Along the double-well axis, for a long lattice depth of $40E_r^{\text{long}}$, the calculated inter-well coupling is 7 Hz; however, effective tunneling is further blocked by local energy detunings arising from spatial inhomogeneity. Crucially, these leakage rates are orders of magnitude smaller than the relevant gate frequencies (typically 1–10 kHz).

Moreover, Fig. S5 shows that the number of "unphysical" qubit states remains nearly unchanged throughout the gate sequence after the initial gate pulse.

Further improvements are straightforward. Initial state preparation with two particles per site on average has already been demonstrated with fidelities exceeding 99% in both bosonic and fermionic systems [6, 32, 31]. Increasing the lattice depth by a factor of two via available laser upgrades would suppress residual tunneling rates by more than two orders of magnitude and further reduce loss channels.

I am not sure if I understand the effect of spatial inhomogeneity on the gate fidelities correctly. From the colormaps (e.g. Figs. 3c & S3a), it looks like the relative frequency error is at the percent level. From the description, it seems to me that this is a coherent effect that could in principle be fixed if one would adjust the hold times for each double-well individually. Assuming that it is just a mismatch of the hold time, a simple fidelity estimate from the oscillations would suggest $F \cos(2 \text{ Pi } J t) = \cos(2 \text{ Pi } \text{eps})$ where eps is relative error on the timing $J t = 1$ arising from a slightly wrong $\Delta J / J = \text{eps}$. With $\text{eps} = 0.01$ this gives $F = 99.8\%$. Do the authors agree with this? It further seems to me that the simulations of the experiment were of the same sort, i.e. including only coherent dynamics with systematic inhomogeneous calibration errors. Is this also correct? The fact that the data shows only Gaussian decay is compatible with only non-Markovian (quasi-static) noise, and makes me wonder about the magnitude of true incoherent (Markovian) noise. In this context, I would appreciate further discussion of fundamental errors sources. Can the authors provide a more complete error model of their device?

(A10) We appreciate this analysis, with which we generally agree. The spatial maps of exchange frequencies indeed indicate that the observed inhomogeneity is a coherent effect that could, in principle, be corrected by adjusting the hold times of each double-well individually or, equivalently, by fine-tuning the local tilt – i.e., flattening the optical potential, which we identify as the dominant source of variation. We note that in the central region used for the quantitative analysis, the relative frequency variation is smaller than 1%. As correctly stated by the referee, our simulations include only coherent dynamics with systematic inhomogeneous calibration errors.

We have not yet reached the stage where we can experimentally characterize the contribution of true incoherent (Markovian) noise, but based on our estimates, it remains negligible in the present regime. Estimates indicate that shot-to-shot fluctuations of the relative lattice phase of about 5 mrad contribute errors on the order of 10^{-5} . The next relevant source is the vacuum lifetime at room temperature, corresponding to a fractional error of 10^{-5} – 10^{-6} per Hubbard tunneling time, which could be further reduced below 10^{-7} in a cryogenic environment. Such cryogenic experiments have so far demonstrated >1 h lifetimes ([37]). Off-resonant photon scattering contributes on the order of 10^{-6} per tunneling

time, and particle loss or tunneling out of the double-well is exponentially suppressed with the lattice depth, currently limited to about 10^{-4} of the gate time. For instance, increasing the long-lattice depth by $9E_r^{\text{long}}$ to $40E_r^{\text{long}}$ improved the ratio of inter- to intra-well tunneling by a factor of eight compared to our previous experiment [5]. Recent unpublished improvements to our apparatus have achieved roughly a factor of two deeper lattices, decreasing the particle loss or tunneling to approximately 10^{-7} of the gate time (as calculated).

Most of these limitations are technical rather than fundamental, and continued engineering improvements can systematically reduce them. The immediate challenges lie in mitigating non-Markovian effects such as optical-potential inhomogeneity, long-term phase drifts between the lattices, and limited experimental duty cycle. Many of these aspects are being directly addressed in our new apparatus dedicated to fermionic quantum computing.

Further concerning the fidelity estimates. The analysis presented in Fig. 4c depends on the initial state. The established standard for estimating gate fidelities, however, is randomized benchmarking (as employed in Ref. 56). An alternative is gate set tomography (as employed in Ref. 57). Why have the authors not performed such a more rigorous analysis? Given the lack of such a benchmarking procedure, I find a direct comparison (of the number 99.75% estimated here) with those of Ref. 56 & 57 unjustified.

(A11) We fully agree that randomized benchmarking and gate set tomography provide the most rigorous assessments of gate fidelities. At the current stage, applying such protocols remains experimentally demanding for our system, but we provide concatenated gate fidelities as viable proxies. Such proxies are also widely used in other platforms and have turned out to be consistent with randomized benchmarking protocols [7]. With the planned implementation of single-qubit control, we will also be able to implement more advanced benchmarking protocols.

In response to the referee's suggestion, we have revised the manuscript to avoid direct numerical comparison with the fidelities reported in Refs. 56 and 57. We also retain the sentence in the conclusion that explicitly acknowledges this planned direction.

RESPONSE TO REPORT FROM THE SECOND REFEREE (B)

We sincerely thank the referee for their careful review and constructive comments. We appreciate their recognition of the broader significance of our work, particularly that collisionally mediated gates with fermionic atoms can provide a direct route towards quantum simulation of fermionic problems, such as determining the electron wave function in molecules and solids. We are encouraged by the referee's positive evaluation of the main advances of our study: the high-fidelity two-atom gates, the composite gate decoupling spin exchange from doublon-hole dynamics, and the demonstration of a long-lived Bell state. We are pleased that these achievements are recognized as a significant step forward for lattice-based quantum simulation.

We also note the referee's request for clarification and additional detail before publication. We are grateful for this opportunity to strengthen the presentation of our results, and we have revised the manuscript accordingly. In the following, we respond point by point to the referee's comments.

Towards this long-term direction, the specific advances reported are (1) the realization of two-atom gates with high fidelity and quality factor; (2) the demonstration of a composite gate that decouples doublon/hole dynamics from spin exchange dynamics; and (3) demonstration of a long-lived Bell state. These advances are significant, and will be considered a significant step forward in the lattice-based quantum simulation.

We thank the referee for this encouraging assessment and are pleased that the advances highlighted are viewed as a meaningful step forward for lattice-based quantum simulation.

Is the 'composite gate' demonstrated here without precedent, or can it be contextualized in prior work? The manuscript was not clear on this point.

(B1) To the best of our knowledge, the implementation of a 'composite gate' in a double-well potential is without precedent in the context of collisional gates. A related concept was proposed only recently in theoretical work [19], underscoring the emerging interest and potential for further progress in this direction. The general principle of "global $N \times \pi/2 \rightarrow$ selective phase shift \rightarrow global $N \times \pi/2$ " sequence is established in quantum computing circuits and originates from standard Ramsey-type protocols.

I found the term "orbital" confusing, as a label of the doublon-hole degree of freedom. "Orbital" physics in optical lattices normally refers to higher-band physics, and is well established as a term. Even in the manuscript, the language shifts to "charge" (instead of "orbital") in later pages. I prefer this – it is unambiguous.

(B2) We have revised the manuscript to consistently use the term "charge" instead of "orbital."

One of the appeals of lattice-based QIP is its scalability. However, the authors demonstrate that fidelity depends sensitively on cloud inhomogeneities. With this perspective, the 99.8% fidelity achieved here is not as impressive as similar fidelities achieved in prior work, for much larger systems (since they were not site resolved). Comparing these two scenarios would be helpful for the reader to contextualize the work. Are inhomogeneities more severe in the author's apparatus, and if so, why?

(B3) We appreciate the opportunity to clarify the significance of the reported fidelity and its implications for scalability. Reducing the infidelity from 7×10^{-3} (best prior bulk measurement) to 2×10^{-3} constitutes a 3.5-fold suppression of residual errors. Crucially, this comparison is conservative: bulk measurements (used for the prior measurement) are inherently less direct, whereas our result is spatially resolved and verifies performance at the individual qubit level. When compared to the previous state-of-the-art in quantum gas microscopes (measured by the same group that originally reported higher bulk fidelities), which are essential for verifying digital circuit performance, our result represents a *22-fold reduction in infidelity*. This very significant increase in performance is largely attributable to our improved optical setup [5, 35], which ensures high phase stability and enables larger lattice constants, a prerequisite for the implementation of future local gates.

Inhomogeneities are indeed more severe in our apparatus due to two reasons: the required imaging resolution of a quantum gas microscope and the light mass of our lithium atoms. Scalability is determined by the trade-off between imaging resolution ($0.82 \mu\text{m}$ at 671 nm in our case) and the lattice spacing required for single-site imaging (here $1.14 \mu\text{m}$). Together with the preparation reservoir, this confines atoms to a circle of $35 \mu\text{m}$ diameter, corresponding to ~ 900 sites in the present setup. Importantly, this is not a hard limit: using higher-numerical-aperture objectives and shorter lattice spacings — for example, $0.3835 \mu\text{m}$ demonstrated in [15] — would increase the number of available sites within the same area to well above 25000. As the lightest alkali atom with a broad Feshbach resonance, lithium enables fast gates but demands deeper lattices and therefore tighter beam waists, which enhances Gaussian curvature and thus sets the current usable system size. This effect is also discussed in [36] (page 38, Fig. 2.4), and several works have demonstrated ways to circumvent Gaussian curvature by using homogeneous lattice beams [34]. One simple way to partially overcome this limitation would be to use diffractive optical elements to homogenize the lattice beam profiles, as demonstrated in [40]. We have clarified the question of scalability in Section 10 of the Methods.

We are aware of the described tradeoffs, and some of them have already been addressed through technical improvements on the new machine, specifically designed for "Fermionic Quantum Processing" with Lithium atoms, which is currently nearing operational readiness. The current results provide a blueprint for achieving high fidelities with a non-negligible number of qubits.

The review of prior work (lines 50-65, where refs 5, 30-38 are discussed) is not clear. It would be helpful to give more specifics – for instance, that all of those prior experimental works concern bosons.

(B4) We have clarified in the introduction that all prior experimental works cited in this section were performed with bosonic atoms.

To further aid comparison and context, the following table summarizes representative collisional gate experiments. The table highlights the progression from early bosonic implementations to the present fermionic realization, which employs lighter atoms, enabling faster dynamics despite the larger lattice constant, and benefits from single-site resolved detection.

Table 1: Comparison of representative collisional gate experiments.

	Mandel et al. [18]	Yang et al. [32]	Zhang et al. [33]	Bojović et al.
Atomic species	^{87}Rb	^{87}Rb	^{87}Rb	^6Li
Lattice constant [nm]	410	767	630	1140
Microscope	No	No	Yes	Yes
Spin-exchange rate [Hz]	~ 6500	709	20.5	1000–13000
Fidelity estimate	55% visibility	99.3%	95.6%	99.75%

Why were the specific magnetic fields (688G, 678G) chosen? Please clarify in the relevant Supp Mat section. Similarly, a gradient (40G/cm) is mentioned on line 277– but this is not meaningful to most readers. Rather, giving the bias in terms of the energy scales of the problem (J or t) would improve clarity.

(B5) The magnetic fields were chosen to maintain the magic ratio during the measurements, as clarified in the Supplementary Information. Regarding the gradient mentioned on line 277, we now provide the conversion to more meaningful energy scales and explicitly give the energy gap between the states $|\uparrow, \downarrow\rangle$ and $|\downarrow, \uparrow\rangle$ in frequency units.

Figure 1a is not needed, nor more clear than the corresponding text without the figure parts.

(B6) We appreciate the referee’s comment. We initially included Fig.1a following the editor’s suggestion. While we do not insist on keeping this figure, we find it helpful for conveying the broader context of the work, particularly regarding the near-term hybrid analog-digital mode. We would respect the editor’s preference on the inclusion of the figure.

I found the sentence spanning lines 328 to 333 unclear. (“Most importantly, the new protocol allows to implement the Z-gate …”) Is this connected to the following sentences? Please improve writing clarity here.

(B7) We thank the referee for bringing this to our attention. We have clarified the wording of the mentioned passage and improved the connection between the sentences. To aid readability, we also revised the corresponding figure and caption.

The "exponential fit" mentioned on line 789 is confusing. The data in Fig S4 appear to be fit with lines, based on their uncertainty bands. Is it more correct to say that a "linear fit" is made, which is interpreted assuming that it is the short-time limit of a simple exponential decay with no offset?

(B8) It is correct that, in the short-time limit, an exponential decay can be well approximated by a linear function, and that this is not readily distinguishable in the plot shown in the paper. For completeness, here we show both the exponential fit used in the manuscript (Fig. R2a) and a corresponding linear fit (Fig. R2b). As expected, both yield essentially identical fidelities. In each case, uncertainties were obtained using bootstrapping, which accounts for the confidence intervals of each data point. We therefore retain the exponential fit in the analysis, as it is the more general and physically motivated description.

Figure R2: Identical extracted fidelities for two fitting methods. We extract fidelities for different post-selected states, illustrated by the cartoons at the top. The solid lines in the corresponding colors show **a**, exponential fits (Fig. S4 of the main text) and **b**, linear fits.

RESPONSE TO REPORT FROM THE THIRD REFEREE (C)

We thank the referee for their careful reading and for their detailed assessment of our work. We appreciate their acknowledgement of the advances achieved with fermionic ${}^6\text{Li}$ atoms, and we value their constructive remarks regarding the challenges of scaling fidelities, mitigating motional effects, and achieving local addressability. The referee has raised several technical and conceptual points, and we appreciate the opportunity to clarify them. Below, we respond to each comment in detail:

In this portion of the discussion I will first discuss spin-exchange entanglement studies, and then pair tunneling and the pair exchange gate. At the end I discuss a number of other topics, such as the lack of individual control in this platform, which will be a large perturbation on the requirements of the platform. I found the spin-exchange portion, which is the first topic presented, and also the highest fidelity, to be the least novel of topics presented in this work. To accomplished with 6Li is an advance that could help with integration, but the physics demonstrated is a repetition of an effect that has been seen again and again starting with the work of Anderlini et al (Ref. [36]). The effect is available for both bosons and fermions, with just a sign of the symmetry changed. In the context of the proposals above, it is also the most ‘replaceable’ in the sense that Rydberg gates could, in principle, be substituted (Ref. [2]). 96% fidelity was previously realized (Ref. 38), with local control in the verification, and this work increases to 99.8%, which is larger, but still not to the level of state of the art neutral atom gates. The coherence achieved of $J \tau_{\text{ex}} / \hbar = 110$ is dominated by inhomogeneities, and the fundamental reason for robustness was already clear in the early work of Ref. [36].

(C1) We agree with the referee that the basic mechanism of collisional gates has been established before [18, 1, 10]. Here, we demonstrate for the first time both high-fidelity and microscopic observation of these interactions, both of which are foundational prerequisites for a fermionic quantum processor. What seems like a modest 4.2% gain in fidelity (95.6% in Ref. [38] \rightarrow 99.8%) is actually a 22-fold reduction in total error, which is a very significant improvement, particularly when considering that the two-qubit gate fidelity enters exponentially upon repeated application in circuits. We therefore see our results as a major accomplishment, as also acknowledged by **referee A** “...experimental results are quite impressive. The claimed high fidelities and long coherence times show how fermionic atoms in optical lattices could become competitive also in a digital setting.” and **referee B** “These advances are significant, and will be considered a significant step forward...”.

Current state-of-the-art neutral atom platforms report two-qubit gate fidelities in the range of 99.4%–99.8% [7, 9, 21, 24, 27], often using fidelity metrics comparable to those employed in this work. Consequently, our measured fidelity of 99.75(6)% places our platform among the highest-performing neutral atom systems to date. Given that little optimization has been incorporated into the optimal control improvements of the gate so far, we believe there is even substantial room for improvement in the future. If the Referee is aware of specific results demonstrating entangling gate fidelities exceeding 99.8%, we would be grateful for the references so that we may update our benchmarking comparison accordingly.

We find that spatial inhomogeneities in the exchange frequency set the dominant decoherence scale arising from the anharmonicity of the optical potential. This reflects not the severity of this mechanism, but rather the fact that other error channels have been reduced to a level below it. Nevertheless, the 110 two-qubit oscillation cycles extracted from our data represent a record for any optical superlattice

platform and a significant achievement even within the broader context of neutral-atom quantum computing (for comparison: ~ 55 coherent cycles in [17]). We are confident that the coherence can be further improved substantially by implementing spin-echo sequences or by employing optical potential flattening techniques, which have already been demonstrated both on our setup and in other quantum gas microscope experiments (Refs. [42-44] in the revised manuscript). Given that there is still considerable room for further technical improvements, we find these initial results extremely encouraging.

As the manuscript progresses into pair tunneling and pair exchange gate, the fundamental sensitivity to motion becomes (even more) important. This is the key challenge in harnessing the native fermionic nature in a variety of protocols. Consequently, the spatial dependence, dephasing, etc. are worse than the spin-exchange case, and represent the key benchmarks for the proposed protocols. For the pair-tunneling gate, I found the (significant) sensitivities and issues well described in the paragraph starting line 186.

(C2) We appreciate that this aspect was found to be clearly described in the manuscript.

For the pair-exchange gate, the authors state that this is a "proof of principle" result, and the fidelity is clearly challenging for the authors. While Fig. 5 provides some context for the overall quality of the gate via the green histograms, I would suggest the authors provide a more compact comparative description of the fidelity. And explain in detail how this fidelity compares to what is required for the proposed protocols.

(C3) The referee is correct in pointing out that the composite pair-exchange gate is more challenging. The green histograms (truth table) give an estimate of the pair exchange fidelity: $F = 0.946(10)$ (not SPAM corrected) and $F = 0.976(10)$ (SPAM corrected). The reduced contrast in measurement in Fig. 5e is primarily caused by direct-exchange processes, which become prominent at the very low lattice depths used there and slightly modify the effective exchange coupling J in the spin and charge sectors. These effects are circumvented at higher lattice depths where direct exchange is negligible. We have clarified this in Sec. 5 of the Methods and updated Fig. 5c and Fig. 5e with a theory curve that accurately captures the observed dynamics.

The fidelity requirements for practical applications of the pair-exchange gate depend strongly on the target algorithms. A widely used framework is the variational quantum eigensolver (VQE), which classically minimizes energy functionals of trial states via gate angles, for example, in quantum chemistry [11]. Owing to its closed-loop structure, VQE can partially compensate for coherent, static errors in its ansatz gates [20], as demonstrated experimentally in Refs. [22, 14], where entangling gates with modest fidelities of about 93% were sufficient to implement VQE for small molecules. We therefore expect that modest improvements in fidelity would enable effective use of the gate in closed-loop VQE optimization, although the precise requirements are the subject of ongoing work.

Other algorithms, such as quantum phase estimation, demand significantly higher fidelities [22] and may require reductions of the full gate error to the 10^{-3} level or below.

As a general comment the phrase ‘quantum chemistry’ is used loosely throughout the paper in the context of a variety of protocols. The mapping between the demonstrations here and quantum chemistry requires a variety of logical steps, and the authors could be more specific in their description.

(C4) Recently, several interesting schemes for fermionic quantum computing with applications in quantum chemistry have been proposed [12, 26, 23], all of which require control over fermionic charge degrees of freedom. The pair exchange gate we realize is specifically relevant for the variational fermionic circuits put forth by Gkritis et al. [11], where it is used to prepare molecular wavefunctions in a superlattice architecture extremely similar to ours.

We have removed several unspecific mentions of quantum chemistry from the main text and made the connection between the pair exchange gate and the proposal [11] more explicit.

I now turn to the topic of local control of the motional gates, which the authors indicate is a clear next step in the conclusion. I found this at odds with a statement earlier in the paper that indicates fidelities can keep being pushed up via for example ‘optical flattening techniques’. While the results here benefit from local measurements, local control is going to be a separate challenge. Single-site control will require a variety of new techniques and concepts that will make tweaks to the fidelities observed here irrelevant.

(C5) The manuscript reports the performance of our platform without employing optimal control, active feedback, or optical potential flattening, which would substantially reduce the dominant sources of decoherence. Even so, the obtained fidelities already reach a level consistent with the progress seen in more established quantum computing platforms, highlighting the promising future potential of this approach.

We agree that local control will be a separate technical challenge and we have already begun working on this on a second experiment. The fidelities demonstrated are independent of this and already now highly relevant, as many analog and hybrid proposals rely on global high-fidelity operations rather than complete local control [25, 19]. In addition, we have developed a digital gate scheme that, to a large extent, is based on the global gates demonstrated here and requires only minimal additional modifications. Moreover, initial demonstrations of local control in related systems have already been achieved [15], building on a series of successful realizations in comparable lattice-based architectures (e.g., [30]), which further supports this direction. Our envisioned approach combines global spin-exchange and pair-tunneling operations with local spin flips and on-site offsets, which can be straightforwardly applied using the DMD already available in our setup. Thanks to the larger lattice spacing, leakage effects are expected to be minimal. Taken together, these considerations ensure that the fidelities extracted here provide a meaningful benchmark for both current and future implementations.

A topic left to the supplement is the pair preparation fraction. This is 60-85% - this number and details of its origin, and impact on operations that do not involve post selection, I believe would best fully described in the main text.

(C6) We appreciate the suggestion. The observed 60–85% range reflects technical aspects specific to operating a large analog quantum simulator, rather than a fundamental limitation. High-fidelity pair preparation is entirely feasible on our platform. Recent work, most notably [31] that achieves 99.3(7)% loading into a low-entropy band insulator, demonstrates that such initialization is readily achievable with the same underlying technology. The preparation fraction only affects how often suitable double-wells are available, and not how well the entangling operations function. Given our aim to keep the main text focused on the central conceptual contributions, we believe the expanded discussion in the new Methods Section 2 — part of the printed Nature article — is the appropriate location for these technical details.

Fig. 5b would be clearer with the very small cartoon at the top better explained.

(C7) We have replaced the figure with a revised version in which the cartoon is larger and more clearly explained. To save space, the previous figure with the experimental protocol has been moved to the SI.

The authors indicate Eq. 2 "closely resembles the structure of a Molmer-Sorensen gate" – the authors should explain more clearly in what way this is key to the physics discussed in the paper.

(C8) We thank the referee for the comment. To avoid confusion by suggesting a deeper analogy than intended, we removed the statement.

References

- [1] Anderlini, M. *et al.* Controlled exchange interaction between pairs of neutral atoms in an optical lattice. *Nature* **448**, 452–456 (2007).
- [2] Benhelm, J., Kirchmair, G., Roos, C. F. & Blatt, R. Towards fault-tolerant quantum computing with trapped ions. *Nature Physics* **4**, 463–466 (2008).
- [3] Bissbort, U. *Dynamical Effects and Disorder in Ultracold Bosonic Matter*. Ph.D. thesis, Universitätsbibliothek Johann Christian Senckenberg (2012).
- [4] Bravyi, S., Gambetta, J. M., Mezzacapo, A. & Temme, K. Tapering off qubits to simulate fermionic Hamiltonians (2017). 1701.08213.
- [5] Chalopin, T. *et al.* Optical Superlattice for Engineering Hubbard Couplings in Quantum Simulation. *Physical Review Letters* **134**, 053402 (2025).
- [6] Chiu, C. S., Ji, G., Mazurenko, A., Greif, D. & Greiner, M. Quantum State Engineering of a Hubbard System with Ultracold Fermions. *Physical Review Letters* **120**, 243201 (2018).
- [7] Evered, S. J. *et al.* High-fidelity parallel entangling gates on a neutral-atom quantum computer. *Nature* **622**, 268–272 (2023).
- [8] Evered, S. J. *et al.* Probing the Kitaev honeycomb model on a neutral-atom quantum computer. *Nature* **645**, 341–347 (2025).
- [9] Finkelstein, R. *et al.* Universal quantum operations and ancilla-based read-out for tweezer clocks. *Nature* **634**, 321–327 (2024).
- [10] Fölling, S. *et al.* Direct observation of second-order atom tunnelling. *Nature* **448**, 1029–1032 (2007).
- [11] Gkritis, F. *et al.* Simulating Chemistry with Fermionic Optical Superlattices. *PRX Quantum* **6**, 010318 (2025).
- [12] González-Cuadra, D. *et al.* Fermionic quantum processing with programmable neutral atom arrays. *Proceedings of the National Academy of Sciences* **120**, e2304294120 (2023).
- [13] Granet, E. *et al.* Superconducting pairing correlations on a trapped-ion quantum computer (2025). 2511.02125.
- [14] Hempel, C. *et al.* Quantum Chemistry Calculations on a Trapped-Ion Quantum Simulator. *Physical Review X* **8**, 031022 (2018).
- [15] Impertro, A. *et al.* Local Readout and Control of Current and Kinetic Energy Operators in Optical Lattices. *Physical Review Letters* **133**, 063401 (2024).
- [16] Leibfried, D. *et al.* Experimental demonstration of a robust, high-fidelity geometric two ion-qubit phase gate. *Nature* **422**, 412–415 (2003).
- [17] Levine, H. *et al.* High-Fidelity Control and Entanglement of Rydberg-Atom Qubits. *Physical Review Letters* **121**, 123603 (2018).

- [18] Mandel, O. *et al.* Controlled collisions for multi-particle entanglement of optically trapped atoms. *Nature* **425**, 937–940 (2003).
- [19] Mark, D. K. *et al.* Efficiently Measuring d-Wave Pairing and Beyond in Quantum Gas Microscopes. *Physical Review Letters* **135**, 123402 (2025).
- [20] McClean, J. R., Romero, J., Babbush, R. & Aspuru-Guzik, A. The theory of variational hybrid quantum-classical algorithms. *New Journal of Physics* **18**, 023023 (2016).
- [21] Muniz, J. A. *et al.* High-fidelity universal gates in the ^{171}Yb ground state nuclear spin qubit (2024). 2411.11708.
- [22] O’Malley, P. J. J. *et al.* Scalable Quantum Simulation of Molecular Energies. *Physical Review X* **6**, 031007 (2016).
- [23] Ott, R. *et al.* Error-Corrected Fermionic Quantum Processors with Neutral Atoms. *Physical Review Letters* **135**, 090601 (2025).
- [24] Radnaev, A. *et al.* Universal Neutral-Atom Quantum Computer with Individual Optical Addressing and Nondestructive Readout. *PRX Quantum* **6**, 030334 (2025).
- [25] Schlömer, H. *et al.* Local Control and Mixed Dimensions: Exploring High-Temperature Superconductivity in Optical Lattices. *PRX Quantum* **5**, 040341 (2024).
- [26] Schuckert, A., Crane, E., Gorshkov, A. V., Hafezi, M. & Gullans, M. J. Fault-tolerant fermionic quantum computing (2025). 2411.08955.
- [27] Senoo, A. *et al.* High-fidelity entanglement and coherent multi-qubit mapping in an atom array (2025). 2506.13632.
- [28] Setia, K., Bravyi, S., Mezzacapo, A. & Whitfield, J. D. Superfast encodings for fermionic quantum simulation. *Physical Review Research* **1**, 033033 (2019).
- [29] Verstraete, F. & Cirac, J. I. Mapping local Hamiltonians of fermions to local Hamiltonians of spins. *Journal of Statistical Mechanics: Theory and Experiment* **2005**, P09012 (2005).
- [30] Weitenberg, C. *et al.* Single-spin addressing in an atomic Mott insulator. *Nature* **471**, 319–324 (2011).
- [31] Xu, M. *et al.* A neutral-atom Hubbard quantum simulator in the cryogenic regime. *Nature* **642**, 909–915 (2025).
- [32] Yang, B. *et al.* Cooling and entangling ultracold atoms in optical lattices. *Science* **369**, 550–553 (2020).
- [33] Zhang, W.-Y. *et al.* Scalable Multipartite Entanglement Created by Spin Exchange in an Optical Lattice. *Physical Review Letters* **131**, 073401 (2023).
- [34] Shao, H.-J. *et al.* Antiferromagnetic phase transition in a 3D fermionic Hubbard model. *Nature* **632**, 267–272 (2024).
- [35] Bourgund, D. Charge correlations in quantum simulation of mixed-dimensional Hubbard systems (2025).

- [36] Koepsell, J. Quantum simulation of doped two-dimensional Mott insulators (2021).
- [37] Schymik, K.-N. *et al.* Single Atoms with 6000-Second Trapping Lifetimes in Optical-Tweezer Arrays at Cryogenic Temperatures. *Physical Review Applied* **16**, 034013 (2021).
- [38] Alam, F. *et al.* Fermionic dynamics on a trapped-ion quantum computer beyond exact classical simulation (2025). 2510.26300.
- [39] Alam, F. *et al.* Programmable digital quantum simulation of 2D Fermi-Hubbard dynamics using 72 superconducting qubits (2025). 2510.26845.
- [40] Wang, Y.-X. *et al.* Homogeneous Fermionic Hubbard Gases in a Flattop Optical Lattice. *Physical Review Letters* **134**, 043403 (2025).

Response to the referee reports

High-fidelity collisional quantum gates with fermionic atoms

Authors: Bojović, Hilker, Wang *et al.*

Dear Federico,

thank you very much for your email and for the positive news. We are delighted that the manuscript “High-fidelity collisional quantum gates with fermionic atoms” is now considered suitable for publication in Nature in principle.

We thank the referees for their careful reading and constructive feedback. We are glad that Referees 1 and 2 now fully support publication, and we have addressed the remaining concerns raised by Reviewer 3 in a final revision, in particular regarding the role of fermionic exchange statistics, the discussion of related bosonic realizations, and the framing of the title.

We will also ensure that the final manuscript is fully in Nature format and ready for publication according to the provided guidelines.

LIST OF CHANGES

- ◇ We shortened the Abstract to follow the guidelines and expanded the discussion of the role of statistics in collisional gates and the discussion of related bosonic realizations.
- ◇ We made minimal cosmetic changes to the figures, such as slight repositioning of subfigures and correction of formatting issues and typos.
- ◇ We split the Supplementary Materials into Methods and Supplementary Information, with the analytical derivation of the Fermi-Hubbard double-well and its figures now included in the Supplementary Information.
- ◇ To accommodate 10 Extended Data figures and tables, we merged previous figures S1 and S2 of the Supplementary Information into a new figure M1 of the Methods, and figures S4 and S6 into a new figure M3. In addition, we merged the tables containing experimental parameters (Tables S1 and S2) into a single table, and Tables S3 and S4 into a new figure M6. All figures convey the same information as before.

RESPONSE TO REPORT FROM THE THIRD REFEREE (C)

We thank the referee for their careful reading and for their detailed assessment of our work. We appreciate their acknowledgement of the advances achieved with fermionic ${}^6\text{Li}$ atoms, and we value their constructive remarks regarding the challenges of scaling fidelities, mitigating motional effects, and achieving local addressability. The referee has raised several important points regarding the interpretation and presentation of our results, and we appreciate the opportunity to clarify them. Below, we respond to each comment in detail:

I thank the authors for the substantive changes made to the manuscript. In particular, the conclusion paragraph beginning 201; Motivated by these prospects; now delineate the steps that exist between the experimental demonstrations in this work and digital fermionic computing with applications in quantum chemistry, although as per some of my original comments, the intermediate steps are multi-fold. The responses to my queries and that of Referee B have clarified some of the relative importance and implications of the spin-exchange and pair-exchange gates. Although it remains that the spin-exchange gate, while improved quantitatively here, is in analogy to multiple prior demonstrations, both in ensemble and particle-resolved versions. The pair-exchange gate is now presented more clearly as a novel protocol with specific implications (eg circa line 470).

My main reservation remains in the role of fermionic nature in the experimental results and the associated manuscript discussion. Both Referee A and myself were surprised to find certain implications in the article and title, given, as the authors state "Our present gates do not yet rely on fermionic exchange statistics"; With this in mind, the authors should (1) specifically discuss this fact in the manuscript (2) continue to discuss bosonic realizations, eg for example leave in the discussion surrounding Refs. [45,61], in the introduction. The authors should also consider whether the title is clearly the right match. In the responses, the authors discuss positive aspects of both the fermionic nature of ${}^6\text{Li}$, as well as the parameters achieved harnessing a low-mass atom.

(C1) We thank the Referee for this constructive comment and for encouraging us to clarify these points in the manuscript.

(1) Role of fermionic exchange statistics. To avoid any confusion regarding the role of fermionic exchange statistics, we have revised the abstract to explicitly state that fermionic exchange statistics will only become relevant once multiple double-wells are coupled. We believe that the revised abstract, together with the concluding paragraph, now clearly places our results on the pathway toward fermionic quantum processing.

(2) Relation to bosonic realizations. Following the Referee's suggestion, we retained the discussion of related bosonic realizations, including references to Refs. [45,61] in the Introduction, to better place our work in the broader context.

Title We also carefully considered the Referee's suggestion regarding the title and believe that it accurately reflects both the experimental results presented here and the longer-term motivation of fermionic quantum computation discussed in the manuscript. In particular, we purposefully refer to fermionic atoms rather than fermionic gates in the title. Together with the revised abstract and the concluding discussion outlining the path toward fermionic quantum computing, we believe this makes clear that fermionic atoms are a central aspect of the work and that the title will not mislead the reader. (We further

hope that the title will stimulate important experimental and theoretical follow-up studies exploring new gate concepts and measurement schemes for Fermi-Hubbard type models, and architectures for quantum information processing and error-correction with fermionic atoms.)